# HSF1 is a driver of leukemia stem cell self-renewal in acute myeloid leukemia

Qianze Dong[1,14], Yan Xiu[1,2,14], Yang Wang[1,14], Christina Hodgson[3], Nick Borcherding[4], Craig Jordan [5], Jane Buchanan[6], Eric Taylor [6], Brett Wagner [7], Mariah Leidinger[8], Carol Holman [8], Dennis J. Thiele[9], Sean O'Brien[9], Hai-hui Xue [10], Jinming Zhao[1,11], Qingchang Li [11], Howard Meyerson[12], Brendan F. Boyce[13] & Chen Zhao [1,2,12] ✉

Acute myeloid leukemia (AML) is maintained by self-renewing leukemic stem cells (LSCs). A fundamental problem in treating AML is that conventional therapy fails to eliminate LSCs, which can reinitiate leukemia. Heat shock transcription factor 1 (HSF1), a central regulator of the stress response, has emerged as an important target in cancer therapy. Using genetic *Hsf1* deletion and a direct HSF1 small molecule inhibitor, we show that HSF1 is specifically required for the maintenance of AML, while sparing steady-state and stressed hematopoiesis. Mechanistically, deletion of *Hsf1* dysregulates multifaceted genes involved in LSC stemness and suppresses mitochondrial oxidative phosphorylation through downregulation of succinate dehydrogenase C (SDHC), a direct HSF1 target. Forced expression of SDHC largely restores the *Hsf1* ablation-induced AML developmental defect. Importantly, the growth and engraftment of human AML cells are suppressed by HSF1 inhibition. Our data provide a rationale for developing efficacious small molecules to specifically target HSF1 in AML.

Acute myeloid leukemia (AML) is the most common acute leukemia in adults and occurs increasingly with age with devastating outcomes[1–3]. Although increasingly refined knowledge of AML biology has led to the development of new targeted agents, such as FLT3, BCL2 or IDH inhibitors, current treatment for most patients still largely relies on standard "7 + 3" chemotherapy and allogeneic stem cell transplantation[2]. The WHO classification divides AML into distinct categories based on recurrent cytogenetic abnormalities, molecular mutations, morphology, and immunophenotypic features. The aberrant leukemia blast proliferation and the stalling of blast differentiation are common features of AML. Regardless of the subtype, all AML is initiated and maintained by small numbers of self-renewing leukemic stem cells (LSCs)[4–7]. A fundamental problem in treating leukemia is that conventional therapy can destroy the bulk of leukemia blasts, but fails to eliminate the LSCs, which can change status and reinitiate malignancy after a period of latency[8,9]. Thus, there is an urgent need to identify and target key molecules that specifically regulate the self-renewal of AML stem cells.

[1]Department of Pathology, Case Western Reserve University, Cleveland, OH 44106, USA. [2]Department of Pathology, Louis Stokes Veterans Affairs Medical Center, Cleveland, OH 44106, USA. [3]MAWD Pathology Group, Kansas City, MO 66215, USA. [4]Department of Pathology and Immunology, Washington University School of Medicine, St Louis, MO 63110, USA. [5]Division of Hematology, University of Colorado Anschutz Campus, Denver, CO 80045, USA. [6]Department of Biochemistry, Carver College of Medicine, University of Iowa, Iowa City, IA 52240, USA. [7]Free Radical and Radiation Biology Program, Department of Radiation Oncology, University of Iowa, Iowa City, IA 52242, USA. [8]Department of Pathology, University of Iowa, Iowa City, IA 52242, USA. [9]Sisu Pharma, Inc., Chapel Hill, NC 27514, USA. [10]Center for Discovery and Innovation, Hackensack University Medical Center, Nutley, NJ 07110, USA. [11]Department of Pathology, China Medical University, 77 Puhe Rd, Shenbei Xinqu, Shenyang Shi 110122 Liaoning Sheng, China. [12]Department of Pathology, University Hospitals Cleveland Medical Center, Cleveland, OH 44106, USA. [13]Department of Pathology and Laboratory Medicine, University of Rochester Medical Center, Rochester, NY 14642, USA. [14]These authors contributed equally: Qianze Dong, Yan Xiu, Yang Wang. ✉e-mail: cxz545@case.edu

AML stem cells are self-renewing quiescent blasts. Depending on the subtype, some LSCs are positive for CD34, like its normal counterpart, but others, for example, AML with NPM1 mutations and AML with monocytic differentiation, are negative for CD34. As there is no general surface marker for LSCs, AML stem cells have recently been enriched through their low reactive oxygen species (ROS) production status compared with the bulk blasts[10]. Increasing evidence has shown that AML stem cells have higher mitochondrial mass than normal hematopoietic stem/progenitor cells (HSPCs). They also have unique mitochondrial characteristics with increased mitochondrial biogenesis that differs from most other cancers in that they are primarily dependent on mitochondrial oxidative phosphorylation (OXPHOS), not glycolysis, to generate ATP for survival[9–11]. Consistent with these observations, blocking mitochondrial oxidative phosphorylation is selectively cytotoxic to AML stem cells[9,11–16].

Heat shock transcription factor 1 (HSF1) upregulates the expression of heat shock proteins (HSPs) and thus protects cells from misfolded protein-induced proteotoxic stress[17–19]. Recent genome-wide studies revealed that HSF1 reprograms the transcription of genes involved in a multitude of processes, including metabolism, gametogenesis, aging, and cancer. The expression, activity, and nuclear localization of HSF1 are generally elevated in cancers in response to the increased biosynthetic demands and oncogenic stresses resulting from rapid cell proliferation and the mutant cancer proteome[18,20–23]. HSF1 plays a fundamental role in cancer biology since deletion of the *Hsf1* gene, or reduction in HSF1 expression, markedly reduces growth, survival, and metastatic potential through the regulation of signal transduction, transcription, translation, apoptosis, mitochondrial metabolism, and ROS clearance in a range of solid tumors[18,24–26]. Although HSF1 is not an oncogene, it enables cancer cells to accommodate imbalances in signaling and alterations in DNA, protein, and energy metabolism, a phenomenon called "non-oncogene addiction"[18,24,27–29].

Emerging evidence supports the involvement of HSF1 in malignant hematopoiesis. *Hsf1* is transcriptionally upregulated by NOTCH1 signaling in T cell acute lymphoblastic leukemia (T-ALL), and conditional, cancer cell-specific *Hsf1* ablation suppressed cell growth using a mouse T-ALL model and human T-ALL cell lines[30]. Although the role of HSF1 in solid cancers has been well characterized and is emerging in leukemia[30–32], to date, few studies have been conducted to specifically evaluate the role and mechanism of action of HSF1 in AML stem cells[33]. Using systemic and conditional *Hsf1* knockout mice, as well as pharmacological inhibition, we show that HSF1 is specifically required for the maintenance of function of both murine and human AML stem cells, while being dispensable for normal steady-state and stressed hematopoiesis. Furthermore, our data suggest that nuclear HSF1 levels may be used as a clinical marker to monitor AML status. Together, these observations strongly support the development of HSF1 inhibitors to target LSC self-renewal for the treatment of patients with AML.

## Results

### HSF1 Is Required for the maintenance of MLL-AF9−induced AML

Given the critical role of HSF1 in solid cancers and its emerging role in leukemia[30–32,34], we investigated the role of HSF1 in AML using conditional *Hsf1* deletion mice in combination with the well-established MLL-AF9−induced AML mouse model[35,36]. We first deleted *Hsf1* in hematopoietic cells by crossing *Hsf1*[fl/fl] mice[37] with Vav-cre mice (hereafter *Hsf1*KO, Supplementary Fig. 1a). In the absence of *Hsf1*, the development of MLL-AF9 leukemia in vivo was significantly delayed (Fig. 1a), suggesting that HSF1 is required for the initiation of MLL-AF9−induced AML. To investigate if HSF1 is required for the maintenance of full-blown AML, which is more important when considering AML patient treatment, we generated conditional *Hsf1* knockout mice by crossing *Hsf1*[fl/fl] mice with ROSA26-CreER[T2] mice (hereafter *Hsf1*[fl/fl]creER). We first established MLL-AF9−immortalized colony-forming

cells through serial plating of MLL-AF9−transduced HSPCs (Lin−cKit+ Sca1+) from *Hsf1*[fl/fl]creER or *Hsf1*[+/+]creER or *Hsf1*[fl/fl] mice. After 4 rounds of serial plating, the MLL-AF9−transformed cells grew in liquid medium in the presence of IL-3 (10 ng/mL) (Fig. 1b). We then plated equal numbers of transformed cells in methylcellulose media in the presence or absence of 4-hydroxytamoxifen (4-OHT) to induce *Hsf1* deletion (Supplementary Fig. 1b). Compared with controls, 4-OHT−induced deletion of HSF1 significantly reduced cell growth and colony formation (Fig. 1c, d). Importantly, the reduced colony-forming ability of 4-OHT−treated leukemia cells lasted after withdrawal of 4-OHT (Fig. 1e).

We transplanted MLL-AF9−transduced HSPCs from *Hsf1*[+/+]creER or *Hsf1*[fl/fl]creER mice into lethally irradiated recipients along with radio-protective/rescue cells. Five weeks later, recipient mice were started on tamoxifen (Tam) treatment by oral gavage (Fig. 1f). Mice that received MLL-AF9−transduced *Hsf1*[+/+]creER cells (treated with Tam), or *Hsf1*[fl/fl]creER cells (vehicle only) developed AML within 3 months. However, the progression of AML in recipient mice receiving MLL-AF9−transduced *Hsf1*[fl/fl]creER cells and treated with Tam was significantly delayed (Fig. 1g). We confirmed that Tamoxifen treatment efficiently deleted *Hsf1* at the genomic level and drove a strong reduction in HSF1 protein abundance (Supplementary Fig. 1c, d). The LSCs in MLL-AF9-induced AML have been defined as lineage marker-negative, cKit-positive, Sca1-negative, CD16-positive, and CD34-positive (defined as lin−ckit+Sca1−CD16+CD34+) granulocyte-macrophage progenitor-like cells[35]. Analysis of mice with fully developed leukemia showed that acute *Hsf1* ablation reduced the LSC frequencies and increased the frequencies of CD11b+ more mature and differentiated leukemia cells. This indicates that HSF1 maintains LSC homeostasis and loss of *Hsf1* accelerates LSC differentiation (Fig. 1h–j). Importantly, when FACS-sorted LSCs from Tam-treated or vehicle-treated (Con) leukemic mice were transplanted into the secondary sublethally irradiated CD45.1 recipients, the repopulating ability of Tam-treated leukemia cells was significantly reduced (Fig. 1k). In addition, when we treated primary recipient mice twice with Tam (with a 10-day interval), no primary and secondary recipient mice developed AML (Fig. 1l, Supplementary Fig. 1e), indicating that *Hsf1* ablation strongly impairs LSC self-renewal capacity. Collectively, these results indicate that HSF1 is required for both the initiation and maintenance of MLL-AF9−induced AML.

### HSF1 is dispensable for steady-state and stressed hematopoiesis

The ideal AML targeting therapy would be specific to AML stem cells, while sparing normal HSPCs. HSPCs are constantly exposed to various physiopathologic stresses, e.g., inflammation, infection, therapeutic drugs, hypoxia, and irradiation. Given the critical role of HSF1 in the maintenance of normal cellular homeostasis and stress resistance in general, we investigated if deletion of *Hsf1* adversely affects normal hematopoiesis using *Hsf1*[fl/fl]Vav-cre mice. Interestingly, deletion of *Hsf1* in all hematopoietic cells had no significant impact on the whole bone marrow (BM) numbers (Supplementary Fig. 2a), lineage distribution (Supplementary Fig. 2b), or the frequencies and absolute numbers of long- and short-term HSCs (Supplementary Fig. 2c, d). In addition, there were no differences in the frequencies of CMP (common myeloid progenitor), GMP (granulocyte-monocyte progenitor), MEP (megakaryocyte-erythrocyte progenitor), and CLP (common lymphoid progenitor) between control and *Hsf1*KO mice (Supplementary Fig. 2e).

As HSF1 is an important regulator of stress responses, we asked if HSF1 is required for stressed hematopoiesis. One exogenous stress frequently used in HSC studies is the treatment of mice with the cell cycle−specific myelotoxic drug, 5-fluorouracil (5-FU). However, we found no differences in survival between *Hsf1*KO and control mice to repeated 5-FU challenges (i.p. 150 mg/kg, on day 0, 7, and 14) (Fig. 2a). In addition, there was no difference in BM HSC number or frequency after sublethal radiation (Fig. 2b) or lipopolysaccharide stimulation (Fig. 2c), two mechanistically distinct stresses compared with 5-FU treatment. BM transplantation, especially serial BM transplantation is

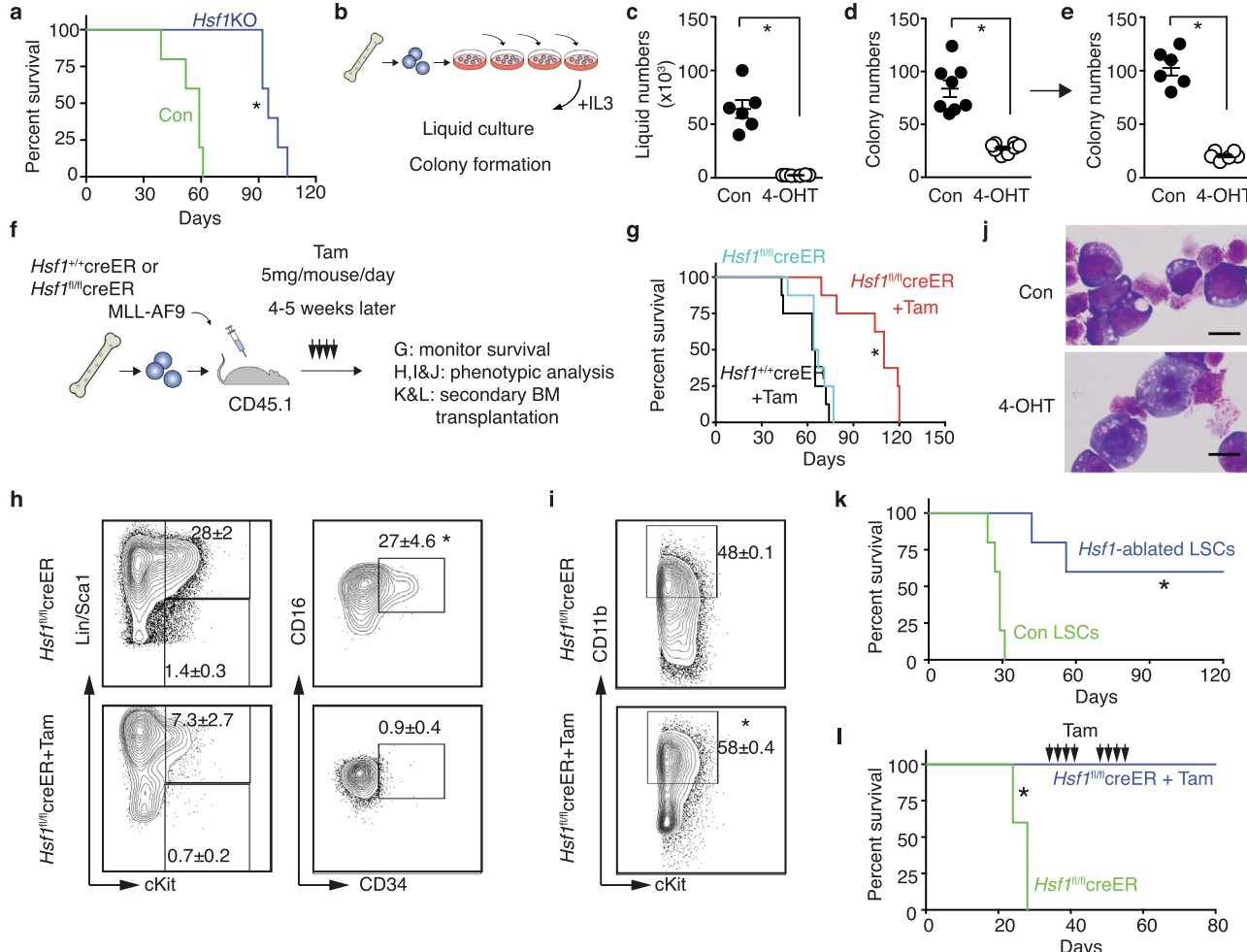

**Fig. 1 | HSF1 is required for the initiation and maintenance of MLL-AF9–induced AML. a** Survival curve of mice receiving MLL-AF9 transduced LSK (Lin⁻cKit⁺Sca1⁺) cells from $Hsf1^{fl/fl}$ (Con) or $Hsf1^{fl/fl}$Vav-Cre + ($Hsf1$KO) mice ($n$ = 5 mice/group). Log-rank test, *$p$ = 0.0021. **b** Scheme of serial plating for establishing immortalized leukemia cells. **c** MLL-AF9 $Hsf1^{fl/fl}$creER leukemia cell numbers in the presence or absence (vehicle, Con) of 4-hydroxytamoxifen (the active metabolite of tamoxifen, 4-OHT, 200 nM). $1 \times 10^3$ leukemia cells were plated and counted 3–4 days later ($n$ = 6 independent replicates). *$p$ = 2.5 × 10⁻⁵. **d** Colony numbers formed in the presence or absence of 4-OHT ($n$ = 8 independent replicates). *$p$ = 7.3 × 10⁻⁶. **e** 500 cells from vehicle (Con) or 4-OHT treated colony assay (**d**) were replated without further 4-OHT treatment ($n$ = 6 independent replicates). *$p$ = 4.24 × 10⁻⁷. **f** Scheme of establishment of MLL-AF9–induced AML and delivery of tamoxifen to induce $Hsf1$ deletion. **g** Survival curve of mice receiving MLL-AF9–transduced LSK cells from $Hsf1^{+/+}$creER or $Hsf1^{fl/fl}$creER mice with or without Tam treatment

($n$ = 8 mice/group). Log-rank test, *$p$ = 0.0002. **h** Frequencies of LSCs in MLL-AF9 $Hsf1^{fl/fl}$creER leukemic mice with or without Tam treatment ($n$ = 4 independent replicates). *$p$ = 2.2 × 10⁻⁵, **$p$ = 0.017. **i** Frequencies of CD11b⁺ leukemia cells in MLL-AF9 $Hsf1^{fl/fl}$creER leukemic mice with or without Tam treatment ($n$ = 3 independent replicates). *$p$ = 2.79 × 10⁻⁵. In (**c–e, h, i**), two-tailed $t$ test was used, data are presented as mean values ± SEM. **j** Wright-Giemsa staining of LSC-derived colony forming cells with or without 4-OHT treatment ($n$ = 3 independent replicates). Note that Tam treatment induced LSC differentiation with reduced nuclear to cytoplasmic ratio. Scale bar, 5 μM. **k** Survival curve of mice receiving LSCs from full-blown MLL-AF9 $Hsf1^{fl/fl}$creER leukemic mice treated with or without Tam ($n$ = 5 mice/group). Log-rank test, *$p$ = 0.0019. **l** Survival curve of mice receiving LSCs from full-blown MLL-AF9 $Hsf1^{fl/fl}$creER leukemic mice treated with two cycles of Tam ($n$ = 5 mice/group). Log-rank test, *$p$ = 0.0031. Source data are provided as a Source Data file.

the gold standard and most rigorous way to test HSC self-renewal in vivo. Thus, we carried out serial BM transplantation to evaluate HSC self-renewal and multi-lineage differentiation potential in vivo. However, no significant differences in donor chimerisms or lineage commitment were detected, even after tertiary transplantation (Fig. 2d, e). In addition, we found there were no significant impacts in donor chimerisms when $Hsf1$ was acutely ablated in adult hematopoiesis by Tam treatment (Supplementary Fig. 2f, g). Collectively, these data and the two recent reports[38,39] demonstrate that HSF1 is not required for either steady-state or stressed hematopoiesis.

## HSF1 DNA binding is required for the regulation of LSC self-renewal

Classically, the multifaceted functions of HSF1 have been ascribed to its impact on transcriptional regulation. However, HSF1 has context-dependent, transcription-independent roles[40–42]. HSF1 contains the highly conserved N-terminal DNA-binding domain (DBD), hydrophobic heptad repeats (HR-A, HR-B and HR-C), and the C-terminal transactivation domain, which is necessary for the transcription of target genes[43,44]. In addition, the regulatory domain (RD), located between the HR-A/B and HR-C domains, is responsible for suppressing HSF1 activity under non-stress conditions (Fig. 3a)[17,45]. To test if HSF1 DNA binding is required for the maintenance of LSC self-renewal, we retrovirally transduced wild-type $HSF1$, a constitutively active (ΔRDT, deletion of RD along with substitution of a hydrophobic amino acid in the HR-C) mutant, and an inactive (R71G, cannot bind to promoter heat shock elements) mutant $HSF1$[46] into MLL-AF9-expressing $Hsf1^{fl/fl}$creER cells and treated them with or without 4-OHT to excise the endogenous $Hsf1$ gene. The wild type and mutant HSF1 protein expression were documented by Western blotting (Fig. 3b). Impaired colony formation

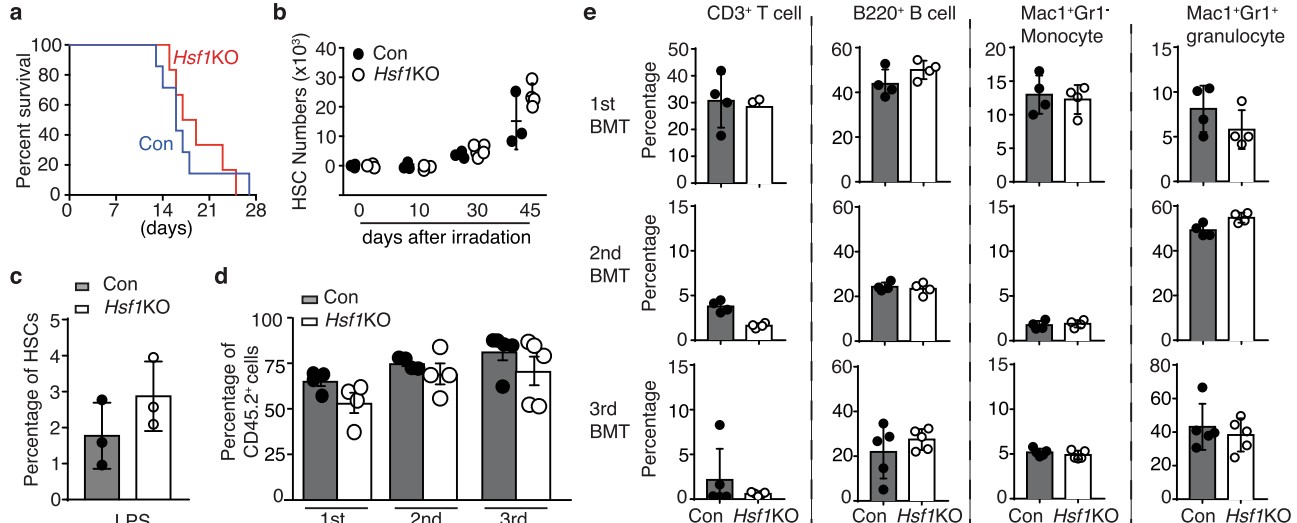

**Fig. 2 | HSF1 is dispensable for stressed hematopoiesis. a** Kaplan–Meier survival curves of *Hsf1*[fl/fl] (Control, Con) or *Hsf1*[fl/fl]Vav-Cre[+] (*Hsf1*KO) mice treated with weekly doses of 5-FU (150 mg/kg, IP, Con n = 7 mice; *Hsf1*KO, n = 6 mice). **b** HSC (defined as Lin[−]cKit[+]Sca1[+]CD150[+] CD48[−]) numbers of Con or *Hsf1*KO mice at different time points after sublethal (6.5 Gy) radiation (Con, n = 3 mice at each time points, *Hsf1*KO, n = 3 mice at Day 0 and 10, n = 5 mice at Day 30 and n = 4 at Day 45). **c** Percentages of HSCs in Con or *Hsf1*KO mice (3 mice/group) treated with 35 μg of LPS twice (48 h apart) and analyzed 24 h after the second injection. **d** Donor chimerisms of mice serially transplanted with control (Con) or *Hsf1*KO HSCs (primary, 500 cells/mouse, secondary and tertiary BMT, 2 × 10⁶ BM cells from primary or secondary recipients. **e** Frequencies of CD3[+] T cells, B220[+] B cells, Mac1[+]Gr1[−] monocytes and Mac1[+]Gr1[+] granulocytes in primary, secondary, and tertiary recipient mice transplanted with Con or *Hsf1*KO KLS cells. **d**, **e** n = 4/group for primary and secondary BMT, and n = 5/group for tertiary BMT. In (**b**–**e**), data are presented as mean values ± SEM. Source data are provided as a Source Data file.

observed for MLL-AF9-expressing *Hsf1*[fl/fl]creER cells in the presence of 4-OHT was partially restored with wild-type HSF1, nearly fully restored with the constitutively active HSF1-ΔRDT mutant and was not improved by expression of the HSF1 R71G mutant that has defective DNA binding (Fig. 3c). These results demonstrate that the DNA binding activity of HSF1 is required for the maintenance of LSC self-renewal and imply that proper HSF1-regulated gene expression, rather than a transcription-independent role, is critical for LSC self-renewal.

### *Hsf1* ablation impairs LSPC proliferation and increases LSPC apoptosis

To evaluate the basis for the dependency of LSPCs (leukemia stem/progenitor cells, defined as CD11b[+]cKit[high+] AML cells[47]) on HSF1, we examined the impact of *Hsf1* deletion on LSPC proliferation, apoptosis, and protein synthesis. *Hsf1* deletion suppressed EdU incorporation (a marker of proliferation) with decreased percentages of LSPCs at S phase and increased percentages of LSPCs at G0/G1 phase compared to control LSPCs (Fig. 3d). In addition, *Hsf1* ablation strongly increased LSPC apoptosis as measured by the elevation in Annexin V and caspase 3 (Fig. 3e, f). As protein translation has been linked to HSF1 function[26], we detected modest, but significantly impaired protein synthesis in the absence of *Hsf1* (Fig. 3g).

### *Hsf1* ablation dysregulates genes involved in multifaceted functions of LSCs

Since our data demonstrate that HSF1 DNA binding is important for LSC self-renewal, we performed RNA-sequencing (RNA-seq, deposited and Source Data) to investigate the underlying molecular mechanisms whereby ablation of *Hsf1* suppresses AML development using FACS sorted *Hsf1*KO or control LSCs. Consistent with HSF1 being a critical regulator of HSPs, the expression of *Hsp90*, *Hspa1a*, *Hspa1b*, *Hspd1*, *Hspe1*, *Hspa4*, and *Hspa8* was significantly reduced in *Hsf1*KO LSCs compared with control LSCs (Fig. 4a). In addition, consistent with the enhanced LSC differentiation and loss of LSC stemness observed in the absence of *Hsf1*, RNA-seq revealed that deletion of *Hsf1* upregulated the expression of genes related to myeloid differentiation: *Itgam/Cd11b*, *Mpo*, *Elane*, and *Csf1*, and downregulated the expression of

genes related to LSC stemness: *Cd34*, *Flt3* and *Bcat1* (Fig. 4b). The upregulation of cell cycle inhibitors Cdkn1a and Cdkn1c, suggested that deletion of *Hsf1* affects cell proliferation (Fig. 4b). Importantly, *Hsf1* ablation downregulated genes related to oxidative phosphorylation (OXPHOS) (Fig. 4c). We also applied Gene Set Enrichment Analysis (GSEA) to identify pathways differentially regulated by *Hsf1* ablation and found that in the absence of *Hsf1*, genes involved in regulating stemness (increased expression of genes that are downregulated in HSCs) (Fig. 4d), OXPHOS (Fig. 4e) and tricarboxylic acid (TCA) cycle (Fig. 4f) were downregulated. These data prompted us to further investigate if HSF1 directly regulates genes involved in mitochondrial metabolism. To this end, we performed CUT&RUN (Cleavage Under Targets & Release Using Nuclease) using sort-purified LSCs. Interestingly, peak call enrichment demonstrated that HSF1-bound genes in LSCs are mainly involved in metabolism, OXPHOS and protein folding (Fig. 4g), consistent with the above RNA-seq results. We focused on one of the direct HSF1 target genes, *Sdhc*, which together with *Sdha*, regulates OXPHOS activity through the electron transport complex II (Fig. 4h, and see below). Collectively, these data suggest that in LSCs, HSF1 regulates a diverse functional array of genes, including mitochondrial metabolism, in addition to its classic role in the heat shock response.

### HSF1 is a key regulator of oxidative phosphorylation in AML

The above data, that *Hsf1* ablation dysregulates genes involved in OXPHOS, prompted us to further investigate the impact of *Hsf1* ablation on AML metabolism, given the fact that AML LSCs are primarily dependent on mitochondrial OXPHOS to generate ATP for survival[11,48,49]. It has been shown that MLL-AF9−induced AML relies on intact glucose metabolism, that LSCs utilize glucose through OXPHOS[50,51], and HSF1 provides sustained glucose uptake by tumor cells[24]. We tested glucose uptake in the presence or absence of *Hsf1* in *Hsf1*[fl/fl]creER LSPCs. However, we found that *Hsf1* ablation had no impact on LSPC glucose uptake, as evaluated by 2-NBDG incorporation (Fig. 5a). We then investigated mitochondrial functions. Although no difference in mitochondrial membrane potential (measured by Mitotracker) was observed between control and *Hsf1*-ablated LSPCs

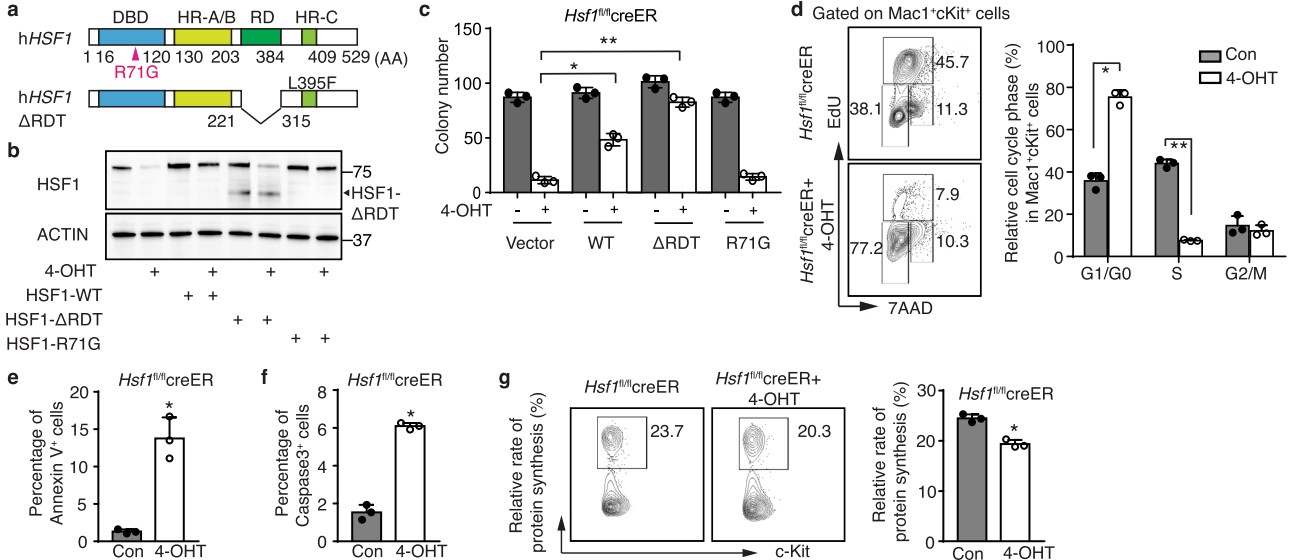

**Fig. 3 | HSF1 DNA binding is required for HSF1 function, and deletion of HSF1 suppresses LSC proliferation and increases LSC apoptosis. a** Scheme of the structure of wild type and mutant alleles of *HSF1*. *DBD*, DNA binding domain; *HR*, hydrophobic heptad repeat; *RD*, regulatory domain. *HSF1* R71G is an inactive and *HSF1ΔRDT* (deletion of regulatory domain and substitution of leucine at amino acid 395 in the suppression domain of the trimerization with glutamic acid, L395F) is an active mutant. **b** Expression of HSF1 and its mutants (*n* = 3 independent replicates). **c** Colony numbers formed using MLL-AF9 *Hsf1*fl/fl creER LSCs transduced with empty vector, *HSF1* WT, inactive R71G mutant, or active ΔRDT mutant in the presence (deletion of endogenous *Hsf1*) or absence of 200 nM 4-OHT (*n* = 3 independent

replicates). \**p* = 0.000605, \*\**p* = 2.39 × 10⁻⁵. **d** EdU incorporation of MLL-AF9 *Hsf1*fl/fcreER LSPCs in the presence of absence of 200 nM 4-OHT (*n* = 3 independent replicates). \**p* = 0.00016, \*\**p* = 5.45 × 10⁻⁶. **e, f** Percentages of Annexin V⁺ (**e**, \**p* = 0.0016) or Caspase-3⁺ (**f**, \**p* = 5.17 × 10⁻⁵) MLL-AF9 *Hsf1*fl/fl creER LSPCs treated with or without 200 nM 4-OHT (*n* = 3 independent replicates). **g** Protein synthesis measured by OP-Puro incorporation in MLL-AF9 *Hsf1*fl/fl creER LSPCs treated with or without 200 nM 4-OHT (*n* = 3 independent replicates). \**p* = 0.00151. In (**c–g**), two-tailed t test was used, data are presented as mean values ± SEM. Source data are provided as a Source Data file.

(Fig. 5b), production of mitochondrial superoxide (MitoSOX) and ROS (measured by H2-DCFDA and CellROX) was significantly increased in *Hsf1*-ablated LSPCs (Fig. 5c–e), which may contribute to the increased apoptosis observed (Fig. 3e–f).

We next measured the oxygen consumption rates (OCRs) and extracellular acidification rates (ECARs) of LSCs directly using a Seahorse Bioscience XF96 extracellular flux analyzer. In this assay, the administration of oligomycin inhibits ATP synthase, resulting in a reduction in mitochondrial respiration or OCR. The injection of Carbonyl cyanide-4 (trifluoromethoxy) phenylhydrazone (FCCP) will enhance electron flow through the electron transport chain (ETC) and drive maximal oxygen consumption by complex IV. A mixture of rotenone (a complex I inhibitor) and antimycin A (a complex III inhibitor) will shut down mitochondrial respiration and enable the determination of the contribution of nonmitochondrial respiration[52]. We found that in the murine MLL-AF9 AML model, MLL-AF9 transduced cKit⁺Lin⁻ cells had higher OXPHOS activity than non-transduced BM cKit⁺Lin⁻ cells (Supplementary Fig. 3a). In addition, *Hsf1* ablation significantly decreased OCR and ECAR in LSPCs (Fig. 5f, g). Consistent with reduced OCR, ATP production from both OXPHOS and glycolysis was reduced by 4-OHT-induced *Hsf1* ablation compared with the control (Fig. 5h). The apparent lack of compensation of ATP production by increased glycolysis in *Hsf1*-ablated LSPCs suggests that LSPCs are in crisis – decreased proliferation (Fig. 3d), increased ROS production (Fig. 5d, e), and increased apoptosis (Fig. 3e). The inability to alter glycolysis to compensate for ETC and TCA cycle dysfunction could also be due to *Hsf1* ablation impairing the expression of genes related to glycolysis (Supplementary Fig. 3b)[53]. To determine how *Hsf1* depletion decreases OXPHOS, we measured changes in metabolite profiles in *Hsf1*fl/fl creER LSPCs 36 h after 4-OHT treatment. Using high-resolution mass spectrometry, we measured cellular metabolites in LSPCs, particularly the TCA cycle intermediates, which supply substrate for OXPHOS directly through the electron transport chain and found that *Hsf1* ablation decreased fumarate and malate levels (Fig. 5i),

consistent with an impaired TCA cycle. We further tested this by tracing the cellular levels of 13C6-glucose-derived metabolites using gas chromatography/mass spectrometry (GC/MS). *Hsf1* ablation reduced the levels of lactate, citrate, glutamate, succinate, malate, and aspartate (Fig. 5j, Supplementary Fig. 3c). These data demonstrate that *Hsf1* ablation impairs ATP production through dysregulation of the TCA cycle at multiple levels, resulting impaired OXPHOS activity.

To identify critical aspects of the TCA cycle directly disrupted by *Hsf1* ablation, we examined expression of key ETC complex I, II, III, IV and V components by Western blotting. *Hsf1* ablation did not alter the protein levels of the key complex I, III, IV, and V subunits, but significantly reduced the complex II components succinate dehydrogenase (SDH) A and SDHC (Fig. 5k). As HSF1 binds *Sdhc* directly (Fig. 4h), downregulation of *Sdhc* was likely regulated at the transcription level by HSF1. qRT-PCR analysis confirmed that the expression of both *Sdha* and *Sdhc* was compromised at the steady state mRNA level in 4-OHT-treated LSCs (Fig. 5l). As succinate is the metabolic substrate and fumarate is the product for ETC complex II, decreased fumarate and malate levels suggested a defect in ETC complex II activity (Fig. 5i, j). Consistent with this prediction, ETC complex II activity was markedly decreased in *Hsf1*-ablated LSPCs compared with control LSPCs (Fig. 5m). These data suggest that SDH is a key enzyme involved in the reduction of OXPHOS caused by HSF1 ablation and is consistent with the strong dependency on the ETC complex II in AML[12,54].

To investigate if downregulation of SDH plays a critical role in *Hsf1* ablation-induced LSC dysfunction, we investigated if overexpression of SDHC could restore defects induced by *Hsf1* ablation. We found that overexpression of SDHC enhanced cell growth and colony formation of *Hsf1*fl/fl creER cells in the presence of 4-OHT in vitro, but overexpression of SDHC did not change the basal level of LSPC growth (Fig. 5n–p). In addition, overexpression of SDHC significantly restored the impaired OXPHOS and the ETC complex II activities (Fig. 5q, r). Importantly, overexpression of SDHC partially accelerated the delayed

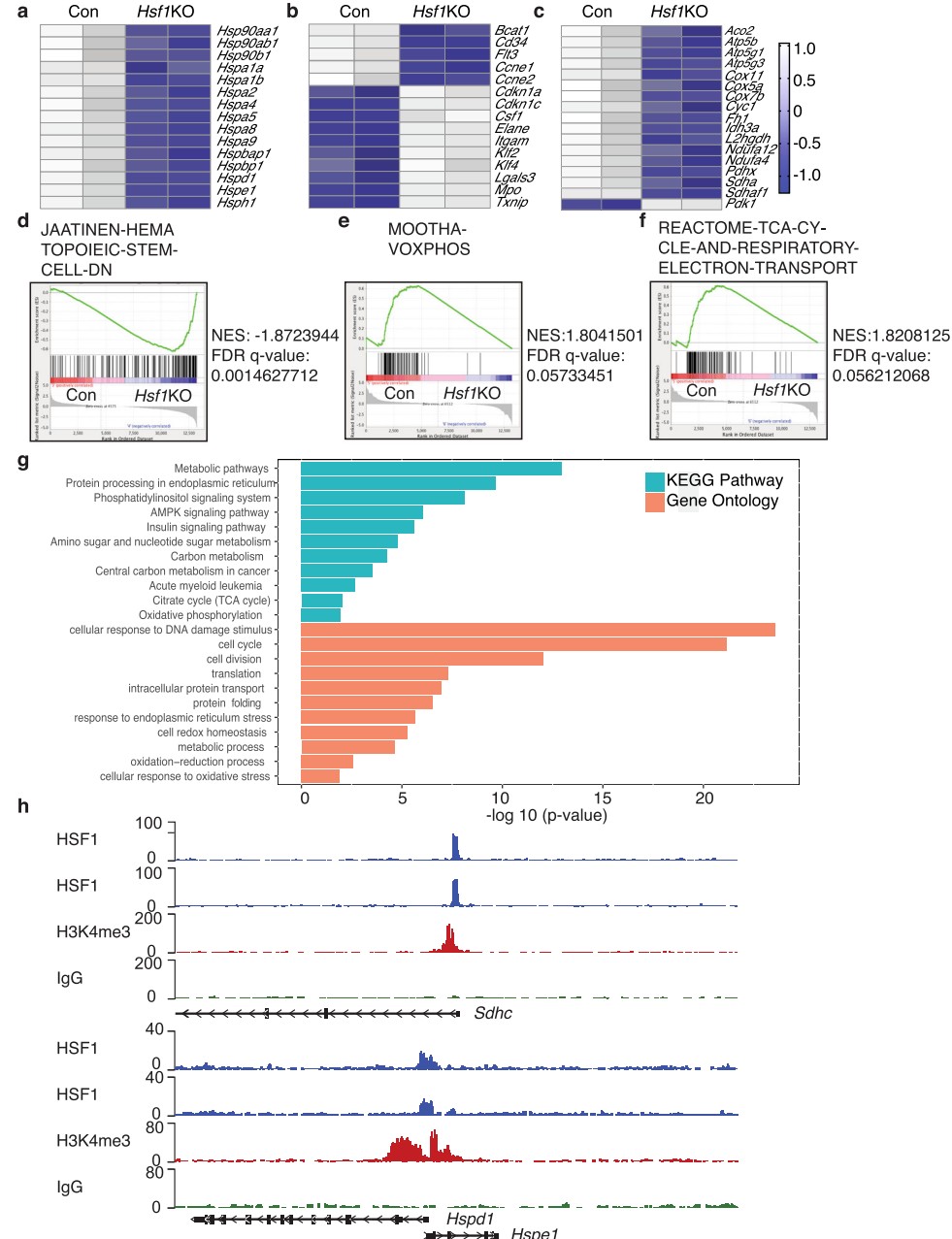

**Fig. 4 | HSF1 regulated genes are involved in multifaceted LSC functions.**
**a–c** RNA-seq analysis of sorted Contol (Con) or *Hsf1*KO MLL-AF9 LSCs (isolated from 2 full blown individual leukemia mice for each genotype). *Hsf1* ablation (**a**) downregulated its known targets—heat shock protein genes; (**b**) upregulated genes related to myeloid differentiation (*Itgam*, *Elane*, *Mpo*) and downregulated stemness related genes (*CD34*, *Flt3*, *Bcat1*), and (**c**) genes related to OXPHOS. **d–f** GSEA revealed that *Hsf1* deletion impaired stem cell signature (**d**), down-regulated genes related to OXPHOS (**e**) and TCA cycle (**f**). **g** CUT&RUN peak call enrichment analyses. KEGG and Gene ontology terms derived from genes involved in top 8000 peaks. **h** CUT&RUN demonstrated HSF1 binds *Sdhc* directly. H3K4me3 shows that HSF1 binding occurs in active chromatin region. Binding to genes encoding heat shock proteins (*Hspd1* and *Hspe1*) and IgG binding were used as positive and negative controls, respectively.

AML development induced by *Hsf1* ablation in vivo (Fig. 5s), suggesting that the HSF1-SDHC axis is an important contributor to LSC self-renewal and AML maintenance.

## HSF1 is not required for human HSPC repopulation, but is critical for LSC self-renewal

To ascertain if HSF1 can be safely targeted in human AML, we first tested if deletion of *HSF1* impacts human HSPC maintenance. Using the CRISPR-Cas9-mediated genome editing system, we efficiently knocked down human *HSF1* in human BM CD34⁺ HSPCs (Fig. 6a). We then transplanted *HSF1* knockout or control human CD34⁺ HSPCs into

sublethally irradiated (2.5 Gy) NSGS (triple transgenic NSG-SGM3 mice expressing human IL3, GM-CSF and SCF) mice. We found that, like *Hsf1* ablation in mice, genetic knockdown of *HSF1* in HSPCs did not affect the BM repopulating capacity of human CD34⁺ HSPCs (Fig. 6b).

Although HSF1 is expressed ubiquitously, *HSF1* transcript expression is significantly lower in normal human CD34⁺ HSPCs than in CD34⁺ leukemic blasts (Fig. 6c). Immunoblotting and immunohistochemistry showed that HSF1 protein is also highly expressed in leukemic blasts, while its expression is very low to negligible in normal BM cells (Fig. 6d, e, Supplementary Fig. 4a). In addition, there is no difference in HSF1 protein expression between non-stem cell

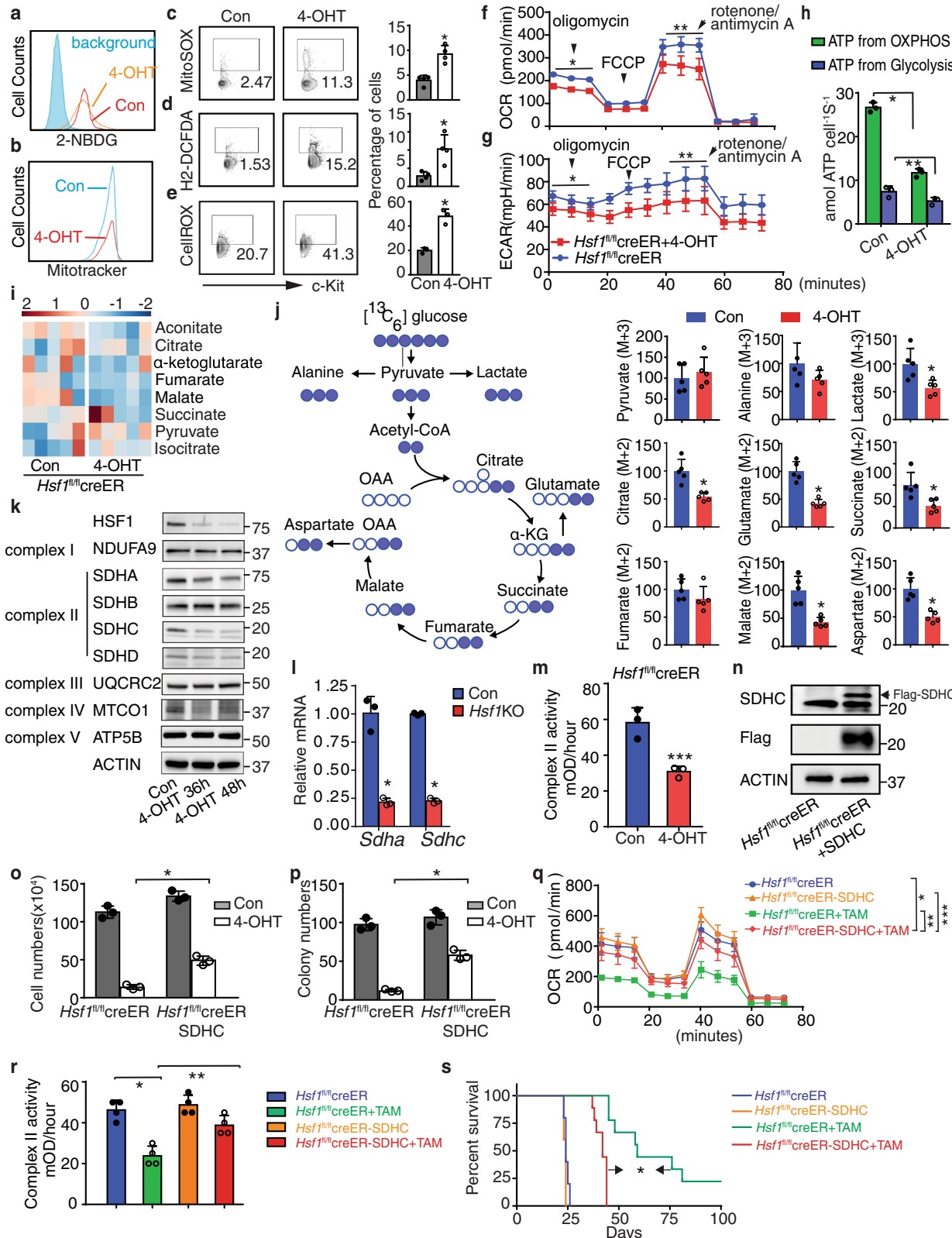

populations and stem cell populations in AML given the diffuse nuclear HSF1 positivity in all AML samples by HSF1 immunostaining (Fig. 6e, Supplementary Fig. 6b). These results not only further substantiate why *HSF1* deletion does not have a significant impact on human HSPC function, but also clearly demonstrate that HSF1 protein is highly expressed in human leukemia cells, with an overall high level in the nucleus by immunohistochemical staining. Consistent with our patient

analysis, analysis of different AML datasets from Oncomine (Supplementary Fig. 4b), TCGA (Supplementary Fig. 4c) and recently published data[55] (Supplementary Fig. 4d, e) show that there is an overall trend that *HSF1* levels are higher in AML patients than healthy controls and especially higher in AML with poor/complex cytogenetics or adverse outcomes. Furthermore, analysis of TCGA data shows that higher *HSF1* expression correlates with a poor prognosis (Fig. 6f).

**Fig. 5 | HSF1 is a key regulator of oxidative phosphorylation in AML.** For experiments (**a**–**m**), $Hsf1^{fl/fl}$creER mouse MLL-AF9 LSPCs were treated with or without (Con) 200 nM 4-OHT for different times. **a** 2-NBDG uptake ($n = 3$ independent replicates). **b** Mitochondrial membrane potential (Mitotracker, $n = 3$ independent replicates). **c** Mitochondrial superoxide (Mitosox, $n = 4$ independent replicates), *$p = 0.0017$. **d**, **e** Mitochondrial ROS levels measured by H2-DCFDA (**d**, $n = 4$ independent replicates, *$p = 0.0106$) or CellROX (**e**, $n = 3$ independent replicates, *$p = 0.0016$). **f**, **g** OCR (**f**, *$p = 5.9 \times 10^{-18}$, **$p = 4.3 \times 10^{-11}$) and ECAR (**g**, *$p = 5.3 \times 10^{-7}$, **$p = 8.8 \times 10^{-8}$) measured by Seahorse ($n = 8$, independent replicates each). **h** ATP production ($n = 3$ independent replicates). *$p = 3.57 \times 10^{-5}$, **$p = 0.000795$. **i** Leukemia cell intermediate metabolites ($n = 5$ independent replicates) using high-resolution Gas chromatograph mass spectrometry (GC/MS). **j** $^{13}$C6-Glucose tracing (measuring leukemia cell intermediate metabolites) using GC/MS. Left, illustration of glucose metabolism. White circles, $^{12}$C carbons; blue circles, $^{13}$C carbons. M + 2 and M + 3 refer to the number of $^{13}$C carbons.; Right, TCA intermediates: citrate (*$p = 0.00167$), succinate (*$p = 0.0121$), fumarate, and malate (*$p = 0.0011$) and intracellular metabolites: pyruvate, alanine, lactate (*$p = 0.0146$), glutamate (*$p = 0.00014$), and aspartate (*$p = 0.00098$). Data are the isotopologue distribution relative to abundance and normalized to the control group, plotted as mean values and individual data points from $n = 5$ cultures. **k** Expression of key electron transport chain complex (ETC) I, II, III, and IV components by Western blotting ($n = 3$ independent replicates). **l** qPCR analysis of expression of $Sdha$ (*$p = 0.0008$) and $Sdhc$ (*$p = 8.2 \times 10^{-7}$) ($n = 3$ independent replicates). **m** ETC complex II activity (*$p = 0.00575$) ($n = 3$ independent replicates). **n** Expression of exogenous SDHC protein in $Hsf1^{fl/fl}$creER LSPCs ($n = 2$ independent replicates). **o**, **p** Leukemia cell proliferation (**o**, *$p = 0.00078$) and colony formation (**p**, *$p = 0.00024$) in empty vector or SDHC transduced $Hsf1^{fl/fl}$creER LSCs in the presence or absence of 200 nM of 4-OHT ($n = 3$ independent replicates, each). **q** OCR in empty vector or SDHC transduced $Hsf1^{fl/fl}$creER LSPCs treated with or without 200 nM 4-OHT ($n = 7$ independent replicates). *$p = 0.00066$, **$p = 2.2 \times 10^{-10}$, ***$p = 4.86 \times 10^{-6}$. **r** ETC complex II activity in empty vector or SDHC transduced $Hsf1^{fl/fl}$creER leukemia cells treated with or without (Con) 200 nM 4-OHT ($n = 4$ independent replicates). *$p = 0.00056$, **$p = 0.0044$. In (**c**–**e**, **h**, **j**, **l**–**m**, **r**), two-tailed t test was used, data are presented as mean values ± SEM. **s** Survival curve of sublethally irradiated recipient mice receiving vector or SDHC transduced $Hsf1^{fl/fl}$creER LSCs treated with vehicle ($n = 5$ mice/group) or TAM ($n = 9$ mice/group). *$p = 0.0001$. Source data are provided as a Source Data file.

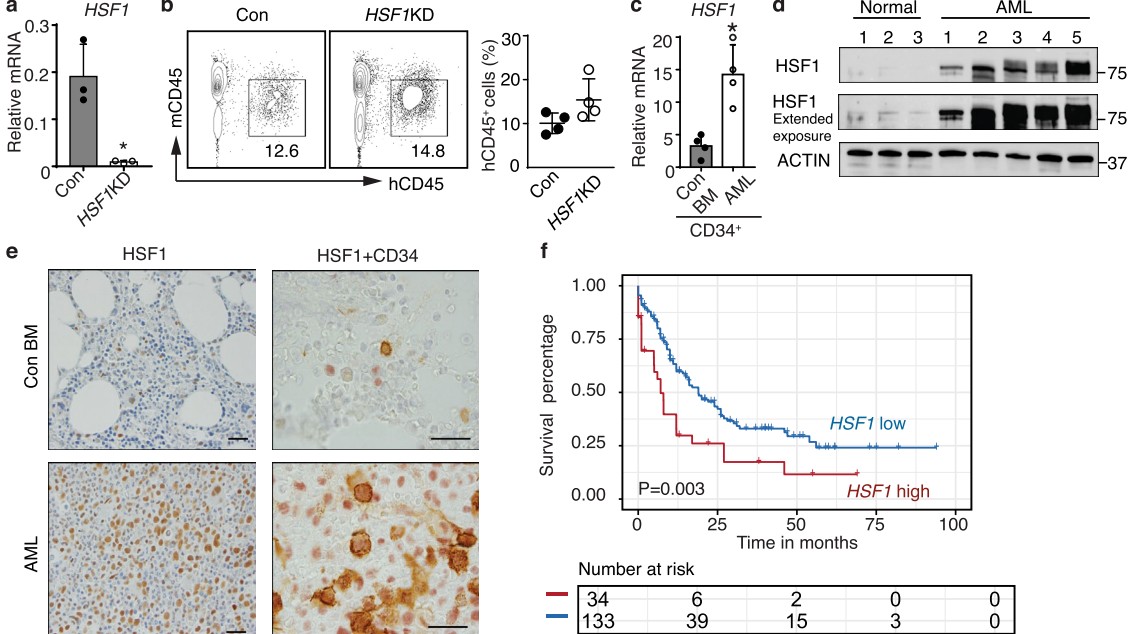

**Fig. 6 | HSF1 is not required for normal human HSPC repopulation but highly expressed in AML cells. a** Relative expression of $HSF1$ mRNA in Cas9 (Con)- or CRISPR-Cas9-mediated $HSF1$ knockdown ($HSF1$KD) in human BM CD34$^+$ HSPCs ($n = 3$ independent replicates). *$p = 0.01017$. **b** Engraftment of Control- or CRISPR-Cas9-mediated $HSF1$ knockdown in human BM CD34$^+$ cells ($n = 4$ mice/group). **c** Relative $HSF1$ mRNA in sorted human healthy BM CD34$^+$ cells and AML CD34$^+$ blasts ($n = 4$ independent replicates). *$p = 0.0041$. In (**a**, **c**), two-tailed t test was used, data are presented as mean values ± SEM. **d** Expression of HSF1 protein in normal (negative lymphoma staging BM, $n = 3$ independent replicates) and primary AML BMs ($n = 5$ independent replicates). Actin is used as a load control. **e** HSF1 staining of HSF1 (nuclear, red) and CD34 (membranous, brown) double-staining of normal and AML BM cells; Scale bar, 20 μM; $n = 16$ primary AML samples. **f** The expression of $HSF1$ correlated with AML patient prognosis (TCGA AML data). Source data are provided as a Source Data file.

We then tested the functional impact of $HSF1$ ablation on the self-renewal and maintenance of human LSCs. Consistent with higher HSF1 expression in leukemic blasts, knockdown of $HSF1$ (70–90% of knockdown efficacy, Supplementary Fig. 5a) significantly reduced human leukemia blast growth in vitro (Supplementary Fig. 5b) and repopulation in NSGS recipient mice (Fig. 7a, see Supplementary Information for limited information on AML samples used for transplantation), ATP production (Fig. 7b) and extended the survival of recipient mice (Supplementary Fig. 5c). Given the dependency of human LSCs on HSF1, we tested if a small molecule HSF1 inhibitor could specifically kill leukemia cells, while sparing normal HSPCs. Previously reported HSF1 pathway inhibitors have been evaluated in cellular and mouse xenograft cancer models[56–58]. However, most either act in an indirect manner in the HSF1 pathway or have an unknown mechanism of action. SISU-102 (recently published as DTHIB, Direct Targeted HSF1 InhiBitor) has been shown to be a direct and selective small-molecule HSF1 inhibitor which physically engages HSF1 and selectively stimulates degradation of the active, homotrimeric, nuclear form of HSF1. SISU-102 robustly inhibits the HSF1 cancer gene signature and drives prostate cancer tumor regression in multiple mouse models[59]. Importantly, SISU-102 administration elicited no observable adverse effects on mouse behavior, weight, or organ histopathology[59]. We first tested the impact of SISU-102 on murine MLL-AF9 leukemia cells and human AML cell lines. As shown in Fig. 7c, SISU-102 clearly inhibited the expression of the HSF1 targets, HSP90 and SDHC, and HSF1 itself, in a dose-dependent manner and strongly suppressed the growth and OCRs of murine MLL-AF9 $Hsf1^{fl/fl}$creER leukemia cells and human MV4–11 and NOMO-1 AML cells (Fig. 7d, e). Importantly,

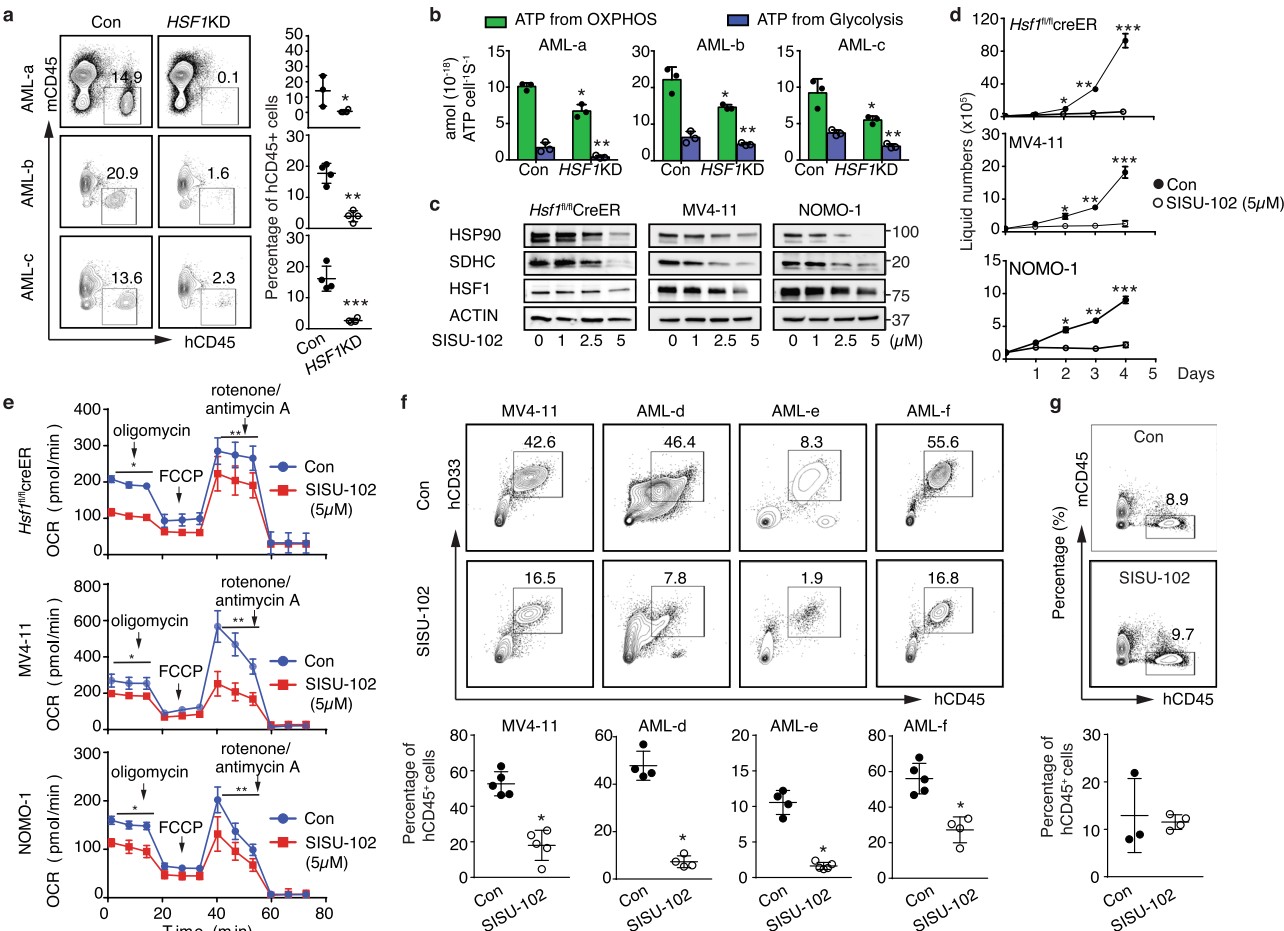

**Fig. 7 | HSF1 is critical for maintenance of LSC self-renewal. a** Engraftment of three different primary AML LSCs in the presence or absence of CRISPR-mediated *HSF1*KD (started 4 mice/AML). *$p = 0.0404$, **$p = 0.0003$, ***$p = 0.0006$. **b** ATP production in control or *HSF1* knockdown AML cells (post-transplantation from **a**) measured by Seahorse assay ($n = 3$ independent replicates). AML-a: *$p = 0.0046$. **$p = 0.0349$; AML-b: *$p = 0.0201$. **$p = 0.0124$; AML-c: *$p = 0.0325$. **$p = 0.0033$; **c** The expression of HSP90, SDHC, and HSF1 proteins in SISU-102 treated mouse MLL-AF9 cells and the human AML cell lines MV4–11 and NOLM-1 ($n = 3$ independent replicates). ACTIN is used as a load control. **d** Cell growth of murine MLL-AF9 cells (*$p = 1.6 × 10^{-5}$, **$p = 3.1 × 10^{-5}$, ***$p = 6.1 × 10^{-5}$) and human AML cell line MV4–11 (*$p = 4.8 × 10^{-3}$, **$p = 2.3 × 10^{-4}$, ***$p = 1.6 × 10^{-4}$) and NOLM-1 (*$p = 6.3 × 10^{-4}$, **$p = 3.01 × 10^{-5}$, ***$p = 6.38 × 10^{-5}$) in the presence of absence of SISU-102 ($n = 3$ independent

replicates). **e** OCR in *Hsf1*fl/fl creER LSCs (*$p = 8.87 × 10^{-31}$, **$p = 1.09 × 10^{-7}$) or human AML cell line MV4–11 (*$p = 4.89 × 10^{-13}$, **$p = 4.97 × 10^{-12}$) and NOLM-1 (*$p = 1.19 × 10^{-18}$, **$p = 0.00028$) treated with or without SISU-102 ($n = 8$ independent replicates). **f** Engraftment of human AML cell line MV4–11 ($n = 5$ mice/group, *$p = 9.8 × 10^{-5}$) or three different primary AML cells in the presence or absence of SISU-102. AML-d: *$p = 1.8 × 10^{-5}$, $n = 4$ mice/group, and AML-e: *$p = 8.98 × 10^{-6}$, Con $n = 4$ mice, SISU-102, $n = 5$ mice; AML-f: *$p = 0.0011$, Con $n = 5$ mice, SISU-102, $n = 4$ mice. **g** Engraftment of human BM CD34[+] HSPCs treated with ($n = 4$ mice) or without ($n = 3$ mice) SISU-102 (5 mg/kg, IP, daily). In (**a, b, d–f**), two-tailed *t* test was used, data are presented as mean values ± SEM. Source data are provided as a Source Data file.

administration of SISU-102 (5 mg/kg; control group injected with proportional amount of DMSO) alone strongly suppressed the engraftment of MV4–11 and three independent primary human AML cells in NSGS recipient mice and extended the survival of recipient mice (Fig. 7f, Supplementary Fig. 5d). Furthermore, the combination of SISU-102 with Ara-C further suppressed the engraftment of human AML cells (Supplementary Fig. 5e). Similar to *Hsf1* genetic knockdown results (Fig. 6b), SISU-102 administration had no overt impact on human BM CD34[+] HSPC engraftment (Fig. 7g). Taken together, these data suggest that pharmacologically targeting HSF1 specifically impairs human LSC self-renewal, while sparing normal HSPC self-renewal.

## Expression of HSF1 serves as a marker to monitor malignant status in AML

In the examination of clinical cases, we frequently observe a hypocellular marrow (5–10% of cellularity) from AML patients at Day14 post-chemotherapy, with no or rare blasts detected by CD34 immunostaining, manual counting and/or flow cytometric analysis. The very low

expression of HSF1 in normal human BMs compared to diffuse positivity of nuclear HSF1 in AML BMs, regardless of CD34 positivity (Fig. 6e), suggests that HSF1 may be more useful than CD34 for monitoring malignant status in AML. To investigate if the expression of HSF1 correlates with AML status, we retrospectively collected 477 cases from 162 patients, 39 of which had serial BM biopsies at diagnosis, remission, and relapse. The H-Score based on HSF1 immunohistochemical staining, calculated by Extent (%) x nuclear intensity (0, 1+, 2+, 3+) of staining showed consistent and statistically significant changes in nuclear HSF1 expression in these serial specimens: high at diagnosis, low at remission, and elevated again at relapse (Fig. 8a, Supplementary Fig. 6a). For example, when the nuclear HSF1 level is low at relapse (Fig. 8b) and further reduced during follow-up (Fig. 8c); this suggests remission; indeed, patients with this HSF1 staining pattern did not experience disease relapse during the follow-up interval. In contrast, during remission, even though the biopsies showed no increase in blasts (by immunohistochemistry, flow cytometry and manual counting), the residual small clusters of HSF1-positive cells (Fig. 8d), although we do not know whether they are residual LSCs with

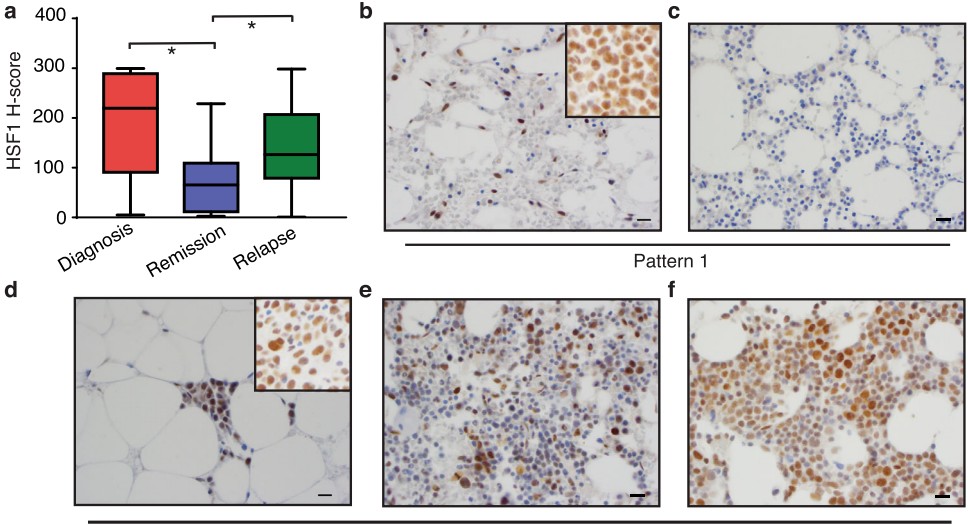

**Fig. 8 | Expression of HSF1 might serve as a marker to monitor AML status.**
**a** HSF1 H-score index in AML samples at diagnosis, remission, or relapse (*n* = 39 primary AML samples). Box plots show median and first/third quartiles. ANOVA with LSD post-hoc analysis was used, *\*p* = 0.0001. **b**–**f** Representative pictures showing 2 different HSF1 staining patterns in AML patients. **b**–**c** Pattern 1,

HSF1 staining at remission (**b**) and follow up (**c**). **d**–**f** Pattern 2, HSF1 staining at remission (**d**), follow up (**e**) and relapse (**f**). Note that (**b**) and (**d**) are 2 weeks post chemotherapy, BM shows 5–10% of cellularity. The insets in (**b**) and (**d**), are HSF1 staining at AML diagnosis. Scale bar, 20 µM. Source data are provided as a Source Data file.

altered immunophenotype or metabolic status due to treatment[8], which persist and are increased in the follow-up (Fig. 8e, f), indicate a risk of relapse, and indeed patient with this HSF1 staining pattern relapsed later. These data suggest that HSF1 nuclear expression might be a biomarker to monitor AML status.

## Discussion

AML remains largely incurable, and it is important to identify new therapeutic targets for AML given the currently limited treatment regimens[2]. Our study demonstrates that HSF1 is specifically required for LSC self-renewal, while being dispensable for normal hematopoiesis. In part, HSF1 maintains LSC self-renewal through the regulation of genes involved in maintaining LSC stemness and ETC complex II-mediated OXPHOS.

Given the critical roles of HSF1 in the maintenance of normal cellular homeostasis[17,19,60], it is somewhat surprising that *Hsf1* ablation has no significant impact on both steady-state and stressed multilineage hematopoiesis. However, this is not completely unpredicted since global *Hsf1*−/− mice have normal development in the absence of acute stress[61,62]. This may be attributable to partial functional redundancy between HSF1 and other HSF family members, such as HSF2, in the regulation of cellular protein homeostasis[63–65]. Indeed, HSF1 and HSF2 can bind to the same cis-acting promoter sequences, and HSF2 activates multiple genes in common with HSF1 in response to febrile stress[64]. In addition, our results are consistent with two recent reports demonstrating that HSF1 is dispensable for normal HSC function, although HSF1 does function in HSC fitness and protein homeostasis in aged mice[30,39]. Importantly, the very low levels of HSF1 protein observed in non-malignant human BM biopsy specimens compared to human AML samples, mouse xenograft experiments using *HSF1* knockdown human BM CD34+ HSPCs, and inhibition of HSF1 function with SISU-102 further support the idea that HSF1 is not required for human HSPC self-renewal. Thus, there is potential for targeting HSF1 in AML.

In contrast, our data clearly demonstrate that HSF1 is required for the maintenance of LSC functions in both MLL-AF9-induced murine and human primary AMLs. It should be noted that the role of HSF1 in LSC maintenance is not limited to AML with MLL rearrangement, as among the six primary AML samples used for xenotransplantation

experiments (Supplementary Table), only one is MLL-rearranged. The generalized HSF1 nuclear positivity in all tested AML samples, regardless of associated cytogenetic/molecular mutations, further support a broader role of HSF1 in AML. Recent studies have shown that AML is primarily dependent on mitochondrial OXPHOS to generate ATP for survival[11,48,49]. Consistent with this, AML blasts have higher mitochondria levels than non-malignant HSCs[49,66]. An increase in mitochondrial mass, mitochondrial membrane potential and OXPHOS predicts AML chemoresistance, while inhibition of OXPHOS can restore sensitivity to chemotherapy[9]. In addition to the classical HSF1 targets, specifically heat shock proteins that function as protein chaperones, the CUT&RUN experiments showed that HSF1 also activates genes involved in AML metabolism. We further determined that *Sdhc* is a direct HSF1 target and *Hsf1* ablation suppressed the expression of SDHC and its family member SDHA, which impaired AML OXPHOS activity. Importantly, overexpression of SDHC partially rescued the LSC functional defects that occur as a consequence of *Hsf1* ablation, although no overt impact on basal LSC growth and engraftment was observed. These results are consistent with the crucial role of HSF1 in regulating mitochondrial metabolism and biogenesis to prevent oxidative damage and increase OXPHOS, which has been reported in other systems[17,67–69], and further supports the notion that ETC complex II activity is essential for LSC survival[70].

Based on the dependence of AML on mitochondrial oxidative phosphorylation, targeting OXPHOS in AML has been proposed as a promising therapeutic approach[15]. Furthermore, the reduced OXPHOS activity in HSF1-ablated LSCs, as well as the multitude of other functions driven by HSF1 that support cancer cell proliferation and survival[18,58,71,72], points to HSF1 inhibition as a potential therapeutic alternative in AML. While targeting HSPs in AML has been explored with promising results[73,74], HSP inhibition interrupts a feedback loop that ultimately increases HSF1 activity and thus compromises treatment efficacy[75,76]. We suggest that targeting HSF1 directly will be more attractive as this strategy can simultaneously downregulate the expression of protein chaperones, impair LSC OXPHOS activity and inhibit a range of oncogenic signaling, survival and metastasis programs that are HSF1-dependent. Moreover, given the high plasticity of AML cells that enables them to adapt their metabolism and ensure their survival during chemotherapy[38], targeting HSF1, either alone or in

combination with conventional agents such as Ara-C could present additional options to target these transient leukemic-regenerating cells[8]. Previous reports suggest targeting HSF1 in AML through microRNAs[77] or via inhibition of translation initiation factor eIF4a[34]. However, directly targeting HSF1 with small molecules such as SISU-102 may have advantages with respect to absorption and pharmaco-kinetic features and would limit potential toxicity from "off target" pathway modulation.

In addition to providing a rationale for targeting HSF1 for AML treatment, our data also suggest that measurement of nuclear HSF1 levels should be further explored as a peripheral biomarker to monitor AML status. Currently, CD34 is the only routinely used clinical marker to enumerate leukemia blasts. However, the blasts in AML with monocytic differentiation (e.g., acute monocytic/monoblastic leukemia, AML with mutated NPM1), acute erythroid leukemia, and acute megakaryocytic leukemia are typically negative for CD34. Monitoring of these leukemias largely relies on morphologic examination, and molecular mutation studies. We provide strong evidence that all tested AML samples, regardless of whether they have recurrent cytogenetic abnormalities or not, expressed higher levels of nuclear HSF1 proteins than normal BM cells, and the expression of HSF1 is not limited to CD34+ blasts. A prospective clinical study using HSF1 immunohistochemical staining, including HSF1 phosphorylation status[78], is warranted. Correlating these characteristics with flow-cytometry and molecular studies could validate the use of HSF1 as a biomarker to predict AML status. In addition, the combination of monitoring HSF1 expression by immunostaining along with HSF1 small molecular inhibitors may be a useful strategy to target AML residual disease just as done for PD-L1 expression with Keytruda and HER2 expression with Trastuzumab in the recently described companion diagnostics[79].

Among the mechanisms that result in increased levels of nuclear HSF1, in a broad range of cancers, include amplification of the *HSF1* gene located within chromosome 8q24.3, reduced expression of the HSF1 degradation F box E3 ligase Fbxw7, activation by loss of p53 and increased *HSF1* transcription by oncogenic signaling pathways such as NOTCH signaling[30,80–82]. Other potential contributing factors are the accumulation of the mutant cancer proteome and rapid proliferation of cancer cells, which may select for increased levels and activity of HSF1 via additional mechanisms. AML is heterogeneous and characterized by a multitude of chromosomal abnormalities and somatic mutations, presenting a requirement for increased levels of the protein quality control machinery to cope with the mutant cancer proteome. Among the many functions of HSF1 in cancer, this increase could, in part, underlie the high dependence on HSF1 for LSC self-renewal. The published AML transcriptome datasets and the large number of AML immunohistochemical staining results demonstrate significantly higher HSF1 expression in different types of AML in general. It should be noted that TP53 is frequently mutated in AML with complex karyotype[83,84], which may account for relatively higher *HSF1* mRNA expression in AML with complex karyotype (Supplementary Fig. 4b, c) given that wild type TP53 suppresses *HSF1* expression and the mutant cancer genome[85]. Thus, targeting a common non-oncogenic and non-essential molecule that is relevant to the progression of different types of AML irrespective of the underlying molecular and cytogenetic aberrations may hold clinical promise. Collectively, these data indicate that HSF1 controls multiple signaling pathways that are required for LSC maintenance, especially OXPHOS metabolism, which is critical for LSC functions, and suggest that targeting HSF1 may have broad anti-leukemic effects.

## Methods

### Mice
The mouse experiments were conducted under approved protocol 2020-0031 of the Institutional Animal Care and Use Committee (IACUC) of Case Western Reserve University. The mice were kept in individually ventilated cages under Specific Pathogen Free conditions at Case Western Reserve University. The mice were fed ad libitum at 24 °C, 50% humidity and a 12:12 h light/dark cycle. The *Hsf1*^fl/fl conditional mouse model has been described previously[37]. The ROSACre-ER^T2 mice were kindly provided by Dr. Yiying Zhang[86] and used previously[87]. Vav-Cre mice (Stock 008610) and the transplant recipient CD45.1 mice (stock 002014) and NSGS mice (stock# 013062) were purchased from Jackson Laboratory. All the mouse strains are in a C57BL6 background and were used at 8–12 weeks old and included male and female mice.

### Human samples and cell lines
De-identified human AML cells were obtained through the CRWU Hematopoietic Biorepository & Cellular Therapy Core (IRB STUDY20210216) and the flow cytometry laboratory at the University of Iowa (Institutional IRB Approval #201508729). The mononuclear cells were purified by Ficoll-density gradient centrifugation and enriched for CD34+ or lineage− leukemic stem/progenitor cells by positive or negative selection using the EasySep™ Human CD34 Positive Selection Kit (#17856) or the Human Progenitor Cell Enrichment Kit (#19356), respectively (StemCell Technologies), depending on CD34 positivity of the blasts. Frozen human BM CD34+ cells (#70002.4) were purchased from StemCell Technologies.

The Human AML cell lines, NOMO-1 (ACC 542) purchased from Leibniz Institute DSMZ-German Collection of Microorganisms and Cell Cultures and MV4–11 (CRL-9591), were purchased from ATCC and cultured in RPMI-1640 media (Gibco) supplemented with 10% FBS (GE Healthcare) and 1% Pen/Strep (Gibco).

### HSPC and LSPC isolation, proliferation, apoptosis and mitochondrial ROS analysis
We used the following antibodies from e-Bioscience and BD Biosciences: B220, CD3, CD4, CD8, CD11b, CD11c, CD16/32, CD34, CD45.1, CD45.2, CD117, Gr-1, Sca1, and Ter119. The antibody was used at a 1:100 for flow cytometry. For isolation of lineage−c-kit+Sca-1+ (KLS) cells, whole BM cells were incubated with a cocktail of lineage antibodies from BD Biosciences (biotinylated anti-mouse antibodies directed against CD3e, CD11b, B220, Gr-1, Ter119) followed by lineage depletion using BD IMag streptavidin particles Plus-DM, then stained with Sca-1 Percp-Cy5.5, CD117-BV421, and streptavidin PE-CF594. For HSPC analyses, lineage depletion was omitted, and the following antibodies were used when appropriate: PE-conjugated lineage markers (CD3, CD4, CD5, CD8, B220, Gr-1, C11b, and Ter119) or biotinylated lineage markers plus streptavidin PE-CF594, Sca-1 PE-Cy5.5, CD117-BV421, CD34-A700 or FITC, CD16/32-PE, CD135-APC, CD127-FITC, or CD150-PECy7, CD48-APC or CD48-PE. For leukemia cell EdU incorporation, proliferation (Ki67+), and caspase3/apoptosis analyses, the following antibodies were used when appropriate: Gr-1-FITC, CD11b-PE or FITC and Sca1-PE or PE-Cy5, CD34-A700, CD117-BV421 or PECy7, CD16/32-APC or -PECy7, CD45.1-APC or A700, CD45.2-PECy7, Ki67-PE, and caspase3-PE.

For in vitro EdU (5-ethynyl-2′-deoxyuridine) labeling, EdU was added to culture media to a final concentration of 10 µM. 1 h later, cells were collected and stained first with surface marker (see above), followed by EdU staining using a EdU-APC staining kit (BD Biosciences). For caspase-3 staining, cells were first stained with surface markers to define the progenitor population, followed by fixation and permeabilization with Cytofix/Cytoperm (BD Biosciences) and intracellular staining with Caspase3-PE. For Annexin V staining, cells were first stained with surface markers to define the progenitor population, washed in 1X binding buffer, and then stained with AnnexinV-APC antibody for 20 min. Cells were resuspended in binding buffer and 5 µl of 7-AAD Staining Solution was added 5 min before analysis using flow cytometry.

For measuring glucose incorporation and mitochondrial ROS, 5 × 10^5 leukemia cells were first stained with Mac1-PE or-APC, c-kit-BV421

or -A700 to define the progenitor population, then labeled with the following agents (all from ThermoFisher Scientific) for 30 min at 37 °C: 2-NBDG (2-(N-(7-Nitrobenz-2-oxa-1,3-diazol-4-yl)Amino)−2-Deoxyglucose, 10 µM, N13195), CellRox (0.75 µM, C10422) or H2-DCFDA (2 µM, D399).

For measuring mitochondrial superoxide and mitochondrial membrane potential, $5 × 10^5$ leukemia cells were labeled with MitoSox (2.5 µM, 15 min, M36008) or Mitrotracker green (5 nM, 30 min, M7514) at 37 °C first, then stained with Mac1-PE or-APC, c-kit-BV421 or -A700 to define the progenitor population.

For cell separation based on endogenous ROS levels, leukemia cells were stained with 8 µM CellRox for 15 min at 37 °C, washed with PBS with 2% FBS, and were collected by cell sorting as described previously[10].

LSRII was used for all the analyses, and FACSAria was used for cell sorting. Data were analyzed using FlowJo (TreeStar).

## Electron transport chain complex II activity assay
MLL-AF9 *Hsf1*^fl/fl^CreER leukemia cells ($5 × 10^5$/well) were treated with 4-OHT (200 nM) for 36 h, samples were prepared, and the enzyme activity was quantified by the colorimetric Complex II Enzyme Activity Microplate Assay Kit according to the manufacturer's protocol (Abcam, ab109908).

## Protein synthesis
For Click-iT Protein Synthesis assay, cells were first stained with surface markers to define the progenitor population, then followed by staining using a Click-iT Plus OPP Alexa Fluor 647 Protein Synthesis Assay Kit (ThermoFisher Scientific).

## Plasmid constructs and virus production
The plasmid, pMSCV-MLL-AF9, was described previously[87]. *Sdhc* cDNA were purchased from Origene (#RC205010), and *HSF1* (human) Wild type, an active (ΔRD, deletion of RD along with substitution of a hydrophobic amino acid in the HR-C L395F), and an inactive (R71G, cannot bind to the heat shock element) mutant *HSF1* were kindly provided by Dr. Nakai[46] and were used as templates for subcloning into BamH1/EcoRI sites of pMSCV-IRES-mCherry using in-fusion following the manufacturer's instructions (Clontech). 293T cells were transiently transfected with MSCV vectors with pCL-Eco or pLV vectors with psPAX2 and pMD2.G using lipofectamine 3000 for retrovirus and lentivirus production, respectively.

## RNA extraction and quantitative real-time PCR analysis
RNA was extracted using a NucleoSpin RNA Kit (Takara). Equal amounts of RNA were used for reverse transcriptase reactions, according to the manufacturer's instructions (SuperScript III First-Strand Synthesis SuperMix, Invitrogen). Quantitative PCR (qPCR) was performed with the iTaq Universal SYBR Green Supermix (Bio-Rad). Gene expression levels were normalized to beta-actin. The primer sequences used are: *Sdha* forward 5′-AAGCTCTTTCCTACCCGATCAC-3′, *Sdha* reverse 5′-AATGCCATCTCC AG TT GTCCT-3′; *Sdhc* forward 5′-TCTTCCCGCTCATGTACCAC-3′, *Sdhc* reverse 5′-GACAACA CAGCA AGAACCACGA-3′; *B2m*/β2-Microglobulin forward 5′- CTG TATGCTATC CAGAAAAC CC-3′, *B2m*/β2-Microglobulin reverse 5′-TCACATGTCTC GAT CCCAGTAG-3′.

## Western blotting
Cells were lysed in 1x lysis buffer (10X RIPA Buffer, Cell Signaling Technologies, #9806S) with protease inhibitors. Subsequently, 50 µg of total cellular lysates were loaded in 10% sodium dodecyl sulfate polyacrylamide gel electrophoresis (SDS-PAGE) gels, electro-transferred onto Nitrocellulose (NC) membrane and immunoblotted with antibodies to HSF1 (ENZO ADI-SPA-901-F, 1:1000 dilution), HSP90 (Cell Signaling Technologies, 4874S, 1:1000 dilution), NDUFA9

(ab14713, 1:800 dilution), SDHA (ab14715, 1:5000 dilution), SDHB (ab14714,1:800 dilution), SDHC (ab155999,1:800 dilution), SDHD (ab189945,1:800 dilution), UQCRC2 (ab14745,1:800 dilution), MTCO1 (ab14705,1:800 dilution), ATP5B (ab14730,1:800 dilution) (all from Abcam), DDK/FLAG (Origene, TA180144, 1:1000 dilution) or beta-actin (Santa Cruz, sc-47778,1:2000 dilution). Membranes were scanned with the ChemiDoc Touch Imaging system (Bio-Rad).

## In vitro colony-forming assays
Fluorescence-activated cell sorting (FACS). FACS-sorted (lin⁻c-Kit⁺Sca1⁻CD16⁺CD34⁺) MLL-AF9 leukemia cells were plated (100–200 cells/96-well plate, 500–2000/24- or 6- well plate) in methylcellulose media (Methocult M3434, StemCell Technologies) in the presence or absence of 4-hydroxytamoxifen (4-OHT, 200 nM, Sigma T-176). For human leukemia cell colony-forming assays, Methocult H4034 was used. Colonies were counted after 5–7 days and replated with the same numbers of cells.

## CRISPR/Cas9-mediated disruption of *HSF1*
CRISPR/Cas9-mediated deletion of *HSF1* was conducted using published protocol[88]. Protospacer sequences for *HSF1* were identified using the CRISPRscan algorithm (www.crisprscan.org)[89]. DNA templates for sgRNAs were made using the protocol described by Li et al[90] using *HSF1* sgRNA primer: taatacgactcactataGGGGACCCTCG TGAGC-GACCgttttagagctagaa. For human primary AML cells or CD34⁺ BM HSPCs, $2 × 10^5$ cells were used per electroporation using the Neon Transfection System 10 µL Kit (Thermo Fisher Scientific, MPK1096). To electroporate Cas9-sgRNA RNPs, 1.5 µg of sgRNA was incubated with 1 µg Cas9 protein (Invitrogen, A36496) for 20 min at room temperature, and electroporated using the following conditions: 1600V, 10 ms, 3 pulses. The electroporated cells were transferred immediately to a 24-well plate containing 0.5 ml of the corresponding growth medium and then incubated for 48 h in a 5% CO2 incubator.

## Seahorse assays
Cells were treated with 4-OHT (200 nM) or SISU-102 (5 µM) for 36 h, then plated in XF96 cell culture microplate (Agilent Technologies, 102417-100). Experiments measuring oxygen consumption (OCR), extracellular acidification rate (ECAR) and total ATP production were performed using a Seahorse XF Cell Mito Stress Test kit, as described previously[91]. Briefly, $1 × 10^5$ cells per well (5 replicates) were seeded in Poly-D-Lysine (Sigma, P6407)-coated 96-well XF96 microplates. Forty-five min prior to analysis, the medium was replaced with Seahorse XF media (Agilent Technologies, 102353-100) and the plate was incubated at 37 °C. For OCR and ECAR, analyses were performed both at basal conditions and after injection of 1 µM oligomycin (Sigma-Aldrich, Cat# 871744), 20 µM FCCP (Sigma-Aldrich, Cat# C2920), 1 µM Antimycin A (Sigma-Aldrich, Cat# A8774), and 10 µM rotenone (Sigma-Aldrich, Cat# R8875). For quantification of ATP production rate from both glycolytic and mitochondrial pathways and ATP rate index, analyses were performed with injection of 15 µM oligomycin (Sigma-Aldrich, Cat# 871744), 5 µM antimycin A (Sigma-Aldrich, Cat# A8774), and 5 µM rotenone (Sigma-Aldrich, Cat# R8875).

## Metabolic profiling for GC-MS
MLL-AF9 *Hsf1*^fl/fl^creER leukemia cells ($1 × 10^6$/samples, 5 replicates) were treated with or without 4-OHT (200 nM) for 36 h and then washed 3 times with ice-cold PBS, 1 time with ice-cold distilled water, and then immediately frozen in liquid nitrogen. Metabolites were extracted using 1 mL of ice-cold methanol/acetonitrile/water (2:2:1) per sample. Samples were then frozen in liquid nitrogen, followed by 10 min sonication. Samples were incubated at −20 °C for 1 h, followed by 10 min centrifugation at 21,000 × g. Supernatants were transferred to auto-sampler vials and dried using a speed-vac. The resulting dried metabolite extracts were derivatized using methoxyamine hydrochloride

(MOX) and N,Obis (trimethylsilyl)trifluoroacetamide (TMS) and examined by gas chromatography-mass spectrometry (GC-MS), as previously described[92,93]. Briefly, dried extracts were reconstituted in 30 µl of 11.4 mg/mL MOX in anhydrous pyridine, vortexed for 5 min, and heated for 1 h at 60 °C. Next, 20 µl of TMS were added to each sample. Samples were vortexed for 1 min and heated for 30 min at 60 °C. Samples were immediately analyzed using GC-MS.

### $^{13}$C-glucose tracing for GC-MS

MLL-AF9 $Hsf1^{fl/fl}$creER leukemia cells ($10 \times 10^6$/samples, 5 replicates) were treated with or without 4-OHT (200 nM) for 36 h. Culture medium was replaced with glucose-free RPMI-1640 with 2000 mg/L $^{13}$C-glucose for 60 min when it will be long enough for glucose to label glycolysis and the TCA cycle above the limit of quantitation but not so long that label mixing confounds interpretation. Following incubation, cells were washed 3 times with ice-cold PBS, 1 time with ice-cold distilled water, and then immediately frozen in liquid nitrogen. Tracing samples were processed using the above method for profiling samples.

### GC-MS instrumentation

GC separation was conducted on a Thermo Trace 1300 GC fitted with a TraceGold TG-5SilMS column. 1 µl of derivatized sample was injected into the GC operating under the following conditions: split ratio = 20-1, split flow = 24 µl/min, purge flow = 5 ml/min, carrier mode = Constant Flow, and carrier flow rate = 1.2 ml/min. The GC oven temperature gradient was as follows: 80 °C for 3 min, increasing at a rate of 20 °C/min to 280 °C, and holding at a temperature at 280 °C for 8 min. For metabolic profiling, metabolites were detected using a Thermo ISQ 7000 mass spectrometer operated from 3.90 to 21.00 min in EI mode (−70eV) using select ion monitoring (SIM). For tracing, metabolites were detected in CI mode using SIM.

### Data analysis

Acquired GC-MS data were processed by Thermo Scientific Trace-Finder 4.1 software, and metabolites were identified based on the University of Iowa Metabolomics Core facility standard-confirmed, in-house library. NOREVA was used for signal drift correction[94]. Data were normalized to total ion signals and MetaboAnalyst 4.0 was used for further statistical processing and visualization[95]. Natural abundance and isotopologue distributions were calculated, as previously described[96].

### Immunohistochemistry and evaluation of HSF1 expression by digital image analysis

A 4-year retrospective search from Oct. 1, 2015, to July 2, 2019, of BM biopsies with AML diagnoses was performed. The electronic medical record was reviewed for each patient and cases were selected at the following time points: diagnosis, post therapy, remission, and relapse. The HSF1 immunohistochemical staining of archived human AML tissue blocks was performed at the University of Iowa under protocol "The role of HSF1 in regulating hematopoietic and leukemic stem cells (IRB # 201903742)". Unstained BM biopsy or clot section slides from patients with AML at diagnosis, remission and relapse or negative lymphoma staging BM or clot section samples (controls) were used for HSF1 (Invitrogen, Cat#MA5-27688, 1:1000 dilution) and/or CD34 (Dako, Cat#M7165, 1:100 dilution) immunohistochemical staining using standard protocols. Briefly, tissue sections were deparaffinized in xylene and rehydrated in graded alcohols. Endogenous peroxidase was blocked with 3% hydrogen peroxide followed by antigen retrieval. Primary and secondary antibodies were incubated for 60 and 30 min, respectively, at room temperature. After wash, Vectastain ABC reagent was applied, followed by Dako DAB Plus (5 min at RT), Dako DAB Enhancer (3 min at RT) and counterstained with Leica/Surgipath Hematoxylin (1 min at RT). HSF1 immunostained-slides were scanned for digital analysis. Digital slides were annotated using the Case Viewer digital image analysis software 2.4. Five high power fields were selected for each slide from regions with the highest concentrations of immune-positive nuclei. A high-power field area was defined as 0.159 mm$^2$. Digital quantification scoring parameters were set to exclude high background intensity and nonspecific staining. Digital scoring parameters included number and proportion (extent %) of nuclei with negative, weak positive, medium positive, and strong positive staining. The H-score, calculated as Extent (%) x nuclear intensity (0, 1+, 2+, 3+) of HSF1 staining, was reported.

### RNA-Seq and transcriptome analysis

Total RNA was extracted from sorted LSCs (lin$^-$c-kit$^+$Sca1$^-$CD16$^+$CD34$^+$) from two full blown leukemia mice developed from MLL-AF9-transduced $Hsf1^{fl/fl}$Vav-Cre HSPCs and two full blown leukemia mice developed from MLL-AF9-transduced $Hsf1^{fl/fl}$ HSPCs (duplicate per genotype), respectively. cDNA synthesis and amplification were performed using the SMARTer Ultra Low Input RNA Kit (Clontech) starting with 20 ng of total RNA per sample, following the manufacturer's instructions. cDNA was fragmented with Q800R sonicator (Qsonica) and used as input for NEBNext Ultra DNA Library Preparation Kit (NEB). Libraries were sequenced on Illumina's HiSeq2000 in single-read mode with a read length of 50 nucleotides producing 60–70 million reads per sample. Sequence data in fastq format were generated using the CASAVA 1.8.2 processing pipeline from Illumina. Differential expression analysis was performed using the DESeq2 v1.30.1 R package[97]. We considered genes differentially expressed between 2 groups of samples when the DESeq2 analysis resulted in an adjusted $p$-value of <0.01 and the log2-fold change in gene expression greater than or equal to 1.5 or less than or equal to −1.5. Gene set enrichment analysis (GSEA) 3.0 was done according to the previous report using the software downloaded from GSEA website (http://software.broadinstitute.org)[98].

### Cleavage under targets & release using nuclease (CUT&RUN) experiments

CUT&RUN was performed using a CUT&RUN Assay Kit (Cell Signaling Technology #86652). Briefly, 80,000 cells were washed in Wash Buffer and bound to 10 µl of activated Concanavalin A beads. Bead-bound cells were incubated with primary antibodies at 4 °C overnight in antibody binding buffer. HSF1 (ENZO, Cat# ADI-SPA-901-F), α-H3K4me3 (Cell Signaling Technology, Cat#C42D8), and Rabbit IgG (Cell Signaling Technology, Cat#DA1E) primary antibodies were used. Then, the cell-bead mixture was washed with Wash Buffer and incubated with Protein A-MNase for 1 h at 4 °C. After washing with Wash Buffer, 2 mM CaCl$_2$ was added to the samples to activate Protein A-MNase digestion for 30 min on ice. The reaction was stopped with the addition of 2x Stop Buffer containing 20 mM EDTA, 0.05% digitonin, 5 mg/ml RNase A, and 2 pg/ml heterologous spike-in DNA. Released chromatin fragments were purified by DNA Purification Buffers and Spin Columns (Cell Signaling Technology, #14209). Libraries were generated using the SMARTer ThruPLEX DNA-seq Kit (TAKARA) and sequenced on the Illumina's HiSeq2000.

### Bone marrow (BM) transplant experiments

For BM transplantation (BMT), sorted lin$^-$kit$^+$Sca1$^+$ cells (2500 cells/recipient, 6 recipients per genotype) from $Hsf1^{fl/fl}$ or $Hsf1^{fl/fl}$VavCre+ mice along with 2–3 × 10$^5$ radioprotective CD45.1 whole BM cells were injected into retro-orbital venous sinuses of lethally irradiated CD45.1 recipients. In our experience, both retro-orbital and tail vein injection work equally well for normal HSPC transplantation. For leukemia cell transplantation (see below), we used tail vein injection to avoid the occasional localized tumor formation in the orbital region caused by the residual AML cells. Beginning 4 weeks after transplantation and continuing for 16 weeks, blood was obtained

from recipients, subjected to ammonium-chloride potassium red cell lysis, and stained with CD45.2 PECy7 and CD45.1 APC along with B220 BV421 and CD3 PE, Mac-1 FITC and Gr1 PECy5 for monitoring donor engraftment and donor-derived lymphoid and myeloid cells. Successful engraftment was defined as the presence of a distinct CD45.2$^+$CD45.1$^-$ population above a background set by parallel analyses of animals transplanted with only competitor cells. For secondary and tertiary transplantation, $3 \times 10^6$ WBM cells from the primary or secondary transplants were injected into lethally irradiated congenic CD45.1 recipients, respectively.

For human HSPC transplantation, sublethally irradiated NSGS mice (Jackson Laboratories, stock# 013062) were injected with human BM CD34$^+$ HSPCs transduced with pL-CRISPR.EFS.GFP or pL-CRISPR.HSF1.EFS.GFP (knockout HSF1) via tail vein in a final volume of 0.2 ml of PBS with 0.5% FBS. Eight weeks later, recipient mice were sacrificed to analyze the engraftment of donor cells.

### In vivo leukemogenesis

FACS-sorted lin$^-$ckit$^+$Sca-1$^+$ BM cells from *Hsf1*$^{fl/fl}$, *Hsf1*$^{fl/fl}$Vav-cre, *Hsf1*$^{fl/fl}$creER, or *Hsf1*$^{+/+}$creER, mice were cultured in Dulbecco Modified Eagle Medium supplemented with 10% FBS, 50 ng/ml SCF, 50 ng/ml TPO, 50 ng/ml FLT3 ligand, and 10 ng/ml IL-3 (all from Peprotech, Rocky Hill, NJ) overnight. Then the cells were plated on virus-loaded retronectin-coated plates for 24 h and spin-infected with viral supernatant supplemented with polybrene (4 mg/ml) at 1000 g for 90 min at room temperature and transplanted retro-orbitally into lethally irradiated (900 cGy, single dose) CD45.1 recipients (45–100 K/recipient) along with $2–3 \times 10^5$ rescue cells or plated into methycellulose media (M3434). After 4 rounds of serial plating, the MLL-AF9–transformed cells grew in liquid medium in the presence of IL-3 (10 ng/mL). Tamoxifen (TAM, 5 mg/mouse/d, Sigma, St. Louis, MO) was administered daily by oral gavage for 4 days at indicated time points post–BM transplantation. The reason we used 4-OHT for in vitro experiments and TAM for in vivo experiments is because TAM can be converted into 4-OHT in the mouse liver but cannot in vitro. In addition, 4-OHT binds to estrogen receptor ~100-fold more efficiently than tamoxifen[99], but is more expensive. For secondary BM transplantations, bulk leukemia cells (300,000/recipient) or sorted leukemia stem cells (500/recipient) from spleens or BM of primary recipient mice were transplanted into sublethally (6.5 Gy) irradiated CD45.1 recipients.

NSGS mice were used for transplantation of MV4–11, primary AML and normal human CD34$^+$ BM cells. NSGS mice were conditioned 24 h prior to transplant with 25 mg/kg busulfan via intraperitoneal (IP) injection. MV4–11 ($5 \times 10^5$ cells/recipient), primary AML cells ($1 \times 10^6$ cells/recipient) or human CD34$^+$ BM cells ($5 \times 10^5$ cells/recipient) in 0.2 ml saline were injected into the tail vein. For delivery of small molecule HSF1 inhibitor, SISU-102, mice were treated with 5 mg/kg SISU-102 daily via intra-peritoneal (IP) injection 5 days after transplantation for 2–3 weeks. SISU-102 (DTHIB) was prepared, as previously described[59]. Donor chimerisms were analyzed 8–10 weeks later using BM cells by flow cytometry after staining with hCD45-APC or-FITC, mCD45.1-A700 or-APC and hCD33-BV421. For all transplanted mice, animals were euthanized if the following symptoms were observed: anemia (pale toes), rough hair coat, hunched backs, weight loss (15% loss than control mice, or breathing problems.

### Statistics and reproducibility

Two-sided Student's *t* test was used for all but survival curve and H-score statistical analyses, and significance was set at $p < 0.05$. Values are mean ± standard error of mean (sem). For Kaplan-Meier survival curves, the log-rank test was used. ANOVA with LSD post-hoc analysis was performed to compare H-scores among pre-treatment, post-treatment, and relapse samples. Differences with p values <0.05 were considered significant.

### Reporting summary

Further information on research design is available in the Nature Research Reporting Summary linked to this article.

### Data availability

The RNA-seq and CUT&RUN data reported in this paper have been deposited in the National Center for Biotechnology Information's Sequence Read Archive (SRA) database with the BioProject ID: PRJNA782637. All the other data are available within the article and its Supplementary Information. The raw numbers for charts and graphs and the uncropped blots are available in the Source Data file. Source data are provided with this paper.

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

## Acknowledgements

The authors thank Heath Vignes and Michael Shey (University of Iowa Flow Cytometry Core facility), and Mike Sramkosk (CWRU Cytometry Core facility) for cell sorting; Dr. Elisabeth Christians (UPMC Univ. Paris 06, CNRS) for providing the *Hsf1*fl/fl mice; Dr. Akira Nakai (Yamaguchi University School of Medicine, Japan) for providing human *HSF1* mutant constructs. This work was supported by the National Institutes of Health/National Cancer Institute (R01CA237006) and the U.S. Department of Veterans Affairs (I01 BX004255), and the Department of Pathology startup funds (Case Western Reserve University and University Hospitals) to C.Z.

## Author contributions

Q.Z.D, Y.X., Y.W., and J.M.Z performed the experiments and analyzed the data. N.B. assisted with RNA-seq and published AML datasets analysis. H.H.X. advised on CUT&RUN assays. Y.W analyzed CUT&RUN data. J.B. and E.T. assisted with the analysis of metabolomic data. B.W. assisted with Seahorse experiments. D.J. T. and S.O. provided HSF1 inhibitor SISU-102

and assisted with manuscript revision. M.L. and C.H. (Christina) assisted with AML sample immunostaining, and C.H. (Christina) performed slides analysis and interpretation. C.H. (Carol), Q.C.L., H.M., and B.F.B. assisted with data analysis, interpretation, and writing. Q.Z.D and C.Z wrote the paper. C.Z. conceived and supervised the overall study.

## Competing interests

D.J.T. and S.O. are employed by and hold equity in Sisu Pharma, Inc. The remaining authors declare no competing interests.
