## [Peer Review File · Nature Communications]

REVIEWER COMMENTS

Reviewer #1 (Remarks to the Author):

Dong and colleagues explored the role of heat shock factor 1 (HSF1) in acute myeloid leukemia (AML) using the classical AML mouse model based on transplantation of retrovirally MLL-AF9 overexpressing bone marrow (BM) cells. Using (conditional) gene ablation they found that Hsf1 inactivation impairs MLL-AF9 immortalization in vitro (colony formation) and leukemia induction (primary) and maintenance (secondary transplants) in vivo (Fig.1). Ablation of Hsf1 did not significantly affect steady-state hematopoiesis; and no significant effect on stress hematopoiesis mediated by 5-FU, sublethal irradiation, LPS exposure or serial transplantation was observed (Fig.2). Structure-functional addback expression experiments revealed that observed effects in MLL-AF9-mediated transformation involves HSF1-regulated gene expression; in addition, they found that HSF1 ablation altered cycle progression associated with limited induction of apoptosis (Fig.3). Expression profiling indicated that HSF1 regulates genes related to metabolism and OXPHOS in particular including Sdhc on which gene promoter region HSF1 seems to bind (Fig.4). Comparative metabolism characterization indicated that Hsf1 ablation impairs ATP production by altering the TCA cycle resulting in reduced OXPHOS activity. Loss of HSF1 reduced electron transport chain complex II components indicating that HSF1-controlled SDH is critical target which was confirmed by rescuing the delayed leukemia induction upon HSF1 ablation (Fig.5). HSF1 ablation did not affect propagation of primary normal human CD34 HSPC but impaired propagation of human LSCs from three patients in NSGS mice. Interestingly, mouse and human MLLr AML cells were sensitive to a small molecule functional HSF1 inhibitor (Fig.6). Finally, immunostaining of primary BM sections suggests that HSF1 could be a general marker of malignancy (Fig.7). Overall, this is a very interesting paper rich of data, well-written and easy to follow. Although some links between HSF1 and AML have been raised earlier, the majority of the finding is truly novel. In addition, there might even some links for clinical translation of the molecular observations. Nevertheless, I have several, though mostly minor points that could further improve the manuscript.

Major points:

1. A large number of in vitro and in vivo experiments using rMLL-AF9 or human MLLr AML cells illustrate the effects of HSF1 for induction and maintenance of the transformed state. Some effects were also shown in human primary AML cells. One wonders, how dependent are the observed effects on the presence of an MLL-rearrangement? HSF1 seems to be generally higher expressed in AML, and mostly in cases of complex karyotypes: are the observed defects eventually linked to genetically unstable situation? Older studies suggested that a MLLr impairs the function of TP53. In fact, there is some recent data (<https://doi.org/10.1038/s41467-021-24064-1>) suggesting that TP53 controls HSF1 expression. The authors could discuss and speculate.

Minor points:

1. Line 130: "no studies have been conducted to..." , this is in part correct, however, that genetic HSF1 inactivation reduced clonogenic activity of putative cancer stem cells has been previously shown (Newman B, et al., Cancer Research, 72:4551-4561, 2012). Maybe you adapt the sentence and add this reference?
2. The authors showed that Hsf1 ablation reduced the frequencies of putative LSC associated with cellular differentiation (Figure 1H-I): did they also check for their clonogenic potential? Will there be more differentiated colonies? How are the cells look morphologically upon Hsf1 inactivation?
3. Fig.1J is not clear: in the text it says that ..when FACS-sorted LSCs from Tam-treated or vehicle treated (Con) leukemic mice were transplanted into..., but the figure only contains 2 lines, +/- TAM? Which population is this now?
4. In Fig. 1K they show that repetitive TAM application abrogated leukemic induction in primary transplanted mice: was the same observed in secondary transplants?
5. They explored the impact of Hsf1 ablation on normal hematopoiesis: in which mice and at which age was this done: Vav-iCre or CreER? The first is constitutive starting around E12.5; any effects might be compensated, whereas the other would allow to show the imminent effects?
6. To activate Cre-ER in vitro the authors used 4-OHT, while in vivo they used TAM, why?
7. They studied HSF1 transcriptional targets in HSF1KO vs control LSC: this is not clear: when were the cells taken, how many mice were studied? Was this from inducible or constitutively HSF1KO

mice?

8. Transcriptome and ChIP-seq raw data needs to be posed into an open repository (like GEO): here the data is presented in a very selective manner.

9. HSF1 knockdown in primary AML (a-c) was performed (Fig.6G). What was the efficacy of the KD? In addition, we do not know anything about the nature of these primary AML samples: do they carry MLLr, and if so, which ones?

10. Line 391-394: In a situation of no detectable blasts, some residual HSF1-positive remained after chemotherapy: what kind of cells are those then? Very immature LSCs?

11. In the reviewer's opinion, the data of Fig.7 appears rather too preliminary to make any strong general statement, so the authors could eventually implement some of the data in a small panel in Fig.6 (and eventually follow-up these findings in more in-depth study?).

Reviewer #2 (Remarks to the Author):

General comments:

It is generally accepted that leukaemic stem cells (LSCs) in acute myeloid leukaemia (AML), and other stem cell driven leukaemias, are of significant clinical importance. In this manuscript the authors describe the role of Heat Shock Transcription Factor 1 (HSF1), a central regulator of the heat-shock response, in AML, using both in vitro and in vivo approaches. The paper is well written and clearly presented. The data are robust in most parts and the topic is of interest to the reader of Nature Communication.

Initially, using conditional HSF1 deletion mice and MLL-AF9-induced AML mouse model the authors demonstrate that genetic deletion of HSF1 impairs leukaemogenesis and LSC function/self-renewal, without significantly affected normal haematopoiesis. Experimental data in this part are convincing.

By performing RNA-sequencing on HSF1 knockout cells, the authors show that HSF1 ablation led to broad gene deregulation, including downregulation of genes involved in the tricarboxylic acid (TCA) cycle and oxidative phosphorylation (OXPHOS), with further work pointing towards downregulation in complex II activity. This part is also convincing, although additional work is needed to strengthen this part (see specific comments below).

Next the authors show that higher HSF1 expression correlates with a poor prognosis in AML and HSF1 knockdown impairs transplantation of human AML cells, with minimum effect on transplantation of normal (CD34+) stem/progenitor cells. Encouraging data was also obtained using pharmacological inhibition HSF1 (using of SISU-102, a previously described HSF1 inhibitor).

Overall, the authors use robust models of AML and present convincing data regarding critical role of HSF1 in leukaemia. The mechanistic data hints that a contributing factor is downregulation of OXPHOS, which is interesting, although additional experiments need to strengthen this part.

Specific comments:

The authors define the AML LCS population in the MLL-AF9 model as lin-ckit+Sca1-CD16+CD34+ (according to Ref 33) and refer to "LSCs" throughout the text. However, it is not clear how enriched this (granulocyte macrophage progenitors-like?) population is for LSCs, which may have implication when this population of is used in in vitro assays, for example in Figures 4-5. Please clarify in text. Also, can the authors experimentally test whether MLL-AF9 expression drives increased OXPHOS in this (or other relevant LCS-enriched) cell population (compared with non-transduced cells)?

Figure 4, D-F: "Con" and "HSFKO" legends on the graph is confusing (wrong?) – there is only one differentially expressed gene list, which is compared with a gene set, so not sure it's accurate to place these two legends on the graph.

Figure 5, J: Why is glucose only traced for 60min? What is the relative contribution of labelled carbons to each metabolite after such a short labelling period? I would suggest the authors also show the labelling fraction of each isotopologue (m+1....m+6) for the selected metabolites as supplementary data.

Figure 5: Can the authors speculate why there is no compensation by increased glycolysis if the main effect of HSF1 ablation is reduction in Complex II activity?

Figure 5: Does overexpression of SDHC rescue OCR? Does overexpression of SDHC rescue Complex II activity?

Figure 6: Please clarify whether engraftment of human cells is measured in blood or bone marrow? Why is there so low transplantation inefficacy of human CD34 (how many cells transplanted?). Do mice transplanted with human AML cells die of leukaemia? Why do authors only rely on engraftment but not survival of mice?

For in vivo experiments, why are cells transplanted via retro-orbital venous sinuses instead of tail vein injection?

Reviewer #3 (Remarks to the Author):

A fundamental problem in cancer therapy in general, and acute myeloid leukemia (AML) in particular, is that although during the conventional therapy, the malignant cells are killed, specific self-renewing cancer stem cells can survive. Accordingly, AML has been shown to be maintained by leukemic stem cells (LSCs), which have a capacity to re-initiate the progression of AML. The goal is to understand the regulatory mechanisms keeping LSCs alive and supporting their self-renewal. In this study, conventional and conditional HSF1 knockout mice were utilized to show that this transcription factor is required for the initiation and maintenance of murine AML, while sparing the steady-state and stressed hematopoiesis. The depletion of HSF1 was shown to dysregulate genes that are involved in modulating LSC stemness and the heat shock response to suppress the mitochondrial oxidative phosphorylation via reduction of enzymes, SDHC and SDHA, that regulate the mitochondrial respiratory chain complex II. SDHC is a direct target of HSF1 and by forcing the expression of SDHC, AML developmental defects caused by HSF1 gene repression were restored. The growth and engraftment of human AML cell lines and primary AML cells were shown to be reduced by HSF1 knockdown or in response to administration of a direct HSF1 small molecule inhibitor (SISU-102). Finally, the expression of nuclear HSF1 was shown to correlate with the AML status of patients and was proposed as a potential clinical marker to monitor the status and progression of AML.

The topic of the study is very timely and important. The obtained results are mostly of high quality, clearly presented and well controlled (for specific comments, see below). The manuscript is very well written, the methods used are many and in good balance, including appropriate mouse and cell models in combination with human patient material. The authors could give some more insight into the mechanism by which the HSF1 inhibitor could specifically be targeted to LSCs so that it would not affect healthy cells. Although they do show that HSF1 depletion does not affect normal hematopoiesis and that normal cells do not express high levels of HSF1, it would be important to show the effect of this inhibitor (SISU-102) on other cell lines as well. Since inactivation of HSF1 has been proposed as a potent treatment for AML already earlier, for example, by targeting HSF1 with specific miRNAs and by inhibiting eIF4a, the authors should discuss in more detail the benefit(s) of using SISU-102 over the previously presented targeting strategies. It would be interesting to know what upregulates HSF1 expression in AML and whether high HSF1 is a driver of the AML phenotype.

Specific comments:

1. Lines 99-101: The authors are not expected to provide detailed description of all the AML

subtypes, but it could be useful to state that the WHO classification divides AML into distinct categories based on the genetic alterations and immunophenotypic markers. The intriguing part is that the stalling of the blast differentiation can be caused by various alterations resulting in diseases heterogeneity. Perhaps it would make the point even stronger, regardless of the subtype, the LSCs are behind the disease in every case.

After that it would be better to start the more detailed description about LSCs (Line 104), which could be extended to cover also other details additional to the mitochondrial specialties. For example, information about the CD-molecules and other markers that are used later on would be really helpful. What markers are specific for the LSCs? What distinguishes the LSCs from other leukemic cells?

2. Line 108: Please add reference to the sentence describing the OXPHOS.

3. Line 135: Clinical markers are not really malignant and AML is already malignant. Rephrase: nuclear HSF1 levels may be used as a clinical marker to monitor AML status.

4. Suggestion for the conclusions on line 134 onwards: The authors could elaborate more on these ideas by thinking about the possibility for companion diagnostics (e.g. similar to PD-L1 and Keytruda, HER2 and Trastuzumab etc.). HSF1 IHC together with SISU-102 as a CDx would be a useful strategy, which could also be added to the Discussion.

5. Line 146: Albeit only HSF1 is in the focus of this study, mammals do have also other HSFs and therefore, the name of the KO mouse should be changed to HSF1KO so that the readers do not accidentally think that you have knocked out all murine HSFs.

6. Line 146: The sentence related to Figure 1A. The figure indicates the survival difference of the mouse lines, but not exactly the differences in the leukemogenesis. Please consider rephrasing.

7. Is the WB for Figure 1C,D,E also? Rephrase or add the information of successful downregulation of HSF1.

8. Line 172: Correct the typo with the CD34+33 and check the superscripts throughout the manuscript.

9. Line 173: Describe the meaning of these markers. CD11b positivity indicates a state where the cell has progressed with specific differentiation trajectory or? The same comment concerns Line 459.

10. Figure S2C is missing (Line 190) due to a mislabeling of the panels in Figure S2. Please, correct.

11. Line 234: What is the difference between LSCs and LSPCs? If none, please harmonize. Also, the previous text gives an impression that CD11b is a marker for more differentiated state. Here it sounds it would be used as a stem cell marker. Please, clarify.

12. Figure 4A-C contain two columns for Con and HSFKO. Is the data in the columns derived from two different mice? This should be indicated. Figure 4 shows the effect of HSF1 depletion on the expression of selected target genes. It would be interesting to see the total number of genes that are affected by HSF1 depletion, which could be added as a supplementary data.

13. Line 266: The authors claim that CUT&RUN identifies HSF1-regulated genes, although the method only detects HSF1-bound sequences. For example, in HS, a subset of HSF1-bound genes is not differentially expressed upon heat shock. Please, correct.

14. In Figure 5C-E, the labels indicating what is measured in Y-axis of the bar graphs are missing. The two flow cytometry plots are presumably from control mice and 4-OHT treated mice, which should be indicated.

Four compounds, including oligomycin, antimycin A, rotenone, and FCCP, are shown in Figure 5F–G. However, the main text does not mention anything about these compounds. Following is mentioned in the Materials and Methods:

“For quantification of ATP production rate from both glycolytic and mitochondrial pathways and ATP rate index, analyses were performed with injection of 15 μ M oligomycin (Sigma-Aldrich, Cat# 871744), 5 μ M antimycin A (Sigma-Aldrich, Cat# A8774), and 5 μ M rotenone (Sigma-Aldrich, Cat# R8875).”

What do these compounds cause in cells? Please, clarify in the main text.

Lines 316-319: “As succinate is the metabolic substrate and fumarate is the product for ETC II decreased fumarate and malate levels suggested a defect in ETC complex II activity (Figure 5I–J). Consistent with this prediction, ETC II activity was markedly decreased in HSF1-ablated LSCs compared with control LSCs (Figure 5M).” What method was used to test this?

Figure legend for Figure 5I mentions n=6, but there are only five columns. Are there five or six replicates?

Figure legend for Figure 5K is missing.

In Figure 5L, red indicates control and blue indicates HSFKO. This must be an error, since the data suggest that HSF1 depletion leads to increased expression of SHD A and SDH C, although the opposite is indicated in the text.

15. Line 344: It is somewhat confusing what cell types are compared in the figure. Is there a difference in HSF1 mRNA and protein expression levels between normal hematopoietic stem and progenitor cells and LSCs? Is there a difference in HSF1 mRNA and protein expression levels between non-stem cell populations in disease and health states? The authors want to emphasize the role of HSF1 in the self-renewing population, but now it remains unclear what populations are compared in Figure 6.

Lines 349-350: “[...] HSF1 protein is highly expressed in human leukemia cells, with an overall high level in the nucleus.” This refers to Figure 6E, but the figure is of low quality, and the low contrast between HSF1 in red and CD34 in brown makes it difficult to determine if there is high nuclear localization. Presenting results from the quantification that was done according to the methods section would resolve this issue.

Lines 381-383: “[...] diffuse positivity of nuclear HSF1 in AML BMs [...]”. This refers to Figure 6E and has the same issue as above. Is there counterstaining in Figure 6E? Please add scale bars. Close-ups would also be helpful. How many patients were analyzed for Figure 6F?

Figure 6A: The title for the Y-axis is “expression”, which is not descriptive enough. Even if it is explained in the figure legend “mRNA expression” and HSF1 as the histogram title would be clearer. Be consistent with the style how to present the data, for example, in Figure 5L the Y axis title is “Relative mRNA levels”.

“As shown in Figure 6I, SISU-102 clearly inhibited the expression of the HSF1 targets, HSP90 and SDHC, in a dose-dependent manner and strongly suppressed the growth and OCRs of murine MLL-AF9 and human MV4-11 cells (Figure 6J–K).”

Fig. 6I–K show three cell lines, HSF1/floxed CreER, MV4-11, and NOMO-1. NOMO-1 is mentioned only in the Materials and Methods, but it should be mentioned also in the main text. It would be also good to remind the reader that HSF1/floxed CreER is a modified MM-AF9 cell line, since the text for Fig. 6I–K mentions only MLL-AF9.

HSF1 levels in response to SISU-102 should be shown in Figure 6I.

16. The results associated with Figure 7 are difficult to comprehend and follow. The results from patient materials (Lines 384-396, Figure 7) mention HSF1 levels in general, but L396-397 suddenly mentions that only HSF1 nuclear expression can be used to monitor AML status. Figure 7A and S4 has "HSF1 H-index" on the Y-axis but the corresponding figure legends calls it "H-score index". Based on the Methods section (Lines 707-714), this should simply be "H-score", and it should be mentioned in the paper what the score is based on (eg. whole cell HSF1 signal or nuclear only HSF1 signal). Based on the Materials section, it seems like the H-score was calculated for nuclear signal. Additionally, Lines 713-714 states "The intensity score, proportion score, H-score and positivity index were reported", but aside from the H-score box plots in Figures 7A and S4, they are not reported in the paper.

Based on this section, it seems the authors have data from these quantifications that indicates HSF1 was nuclear, so this quantification (cytoplasmic vs. nuclear) should be included in the manuscript. The finding that nuclear HSF1 expression can be used to monitor AML status is a major result in this study, and therefore, it is important to provide as strong evidence for this statement as possible.

Figure 7: Please indicate scale bars either in every figure or collectively in the figure legend. The legend states that the insets in Figure 7B, D, and G are HSF1 stainings. The inset in panel D appears to be CD34 (membraneous staining pattern), perhaps there is a mix up? It would be helpful to present such FFPE sections, where the amount of the tissue is comparable. Now, especially Figure 7D and G contain only a few cells. Is there counterstaining in Figure 7E? The brightness of the images is variable.

RESPONSE TO REVIEWERS' COMMENTS

Reviewer #1:

Dong and colleagues explored the role of heat shock factor 1 (HSF1) in acute myeloid leukemia (AML) using the classical AML mouse model based on transplantation of retrovirally MLL-AF9 overexpressing bone marrow (BM) cells. Using (conditional) gene ablation they found that Hsf1 inactivation impairs MLL-AF9 immortalization in vitro (colony formation) and leukemia induction (primary) and maintenance (secondary transplants) in vivo (Fig.1). Ablation of Hsf1 did not significantly affect steady-state hematopoiesis; and no significant effect on stress hematopoiesis mediated by 5-FU, sublethal irradiation, LPS exposure or serial transplantation was observed (Fig.2). Structure-functional addback expression experiments revealed that observed effects in MLL-AF9-mediated transformation involves HSF1-regulated gene expression; in addition, they found that HSF1 ablation altered cycle progression associated with limited induction of apoptosis (Fig.3). Expression profiling indicated that HSF1 regulates genes related to metabolism and OXPHOS in particular including Sdhc on which gene promoter region HSF1 seems to bind (Fig.4). Comparative metabolism characterization indicated that Hsf1 ablation impairs ATP production by altering the TCA cycle resulting in reduced OXPHOS activity. Loss of HSF1 reduced electron transport chain complex II components indicating that HSF1-controlled SDH is critical target which was confirmed by rescuing the delayed leukemia induction upon HSF1 ablation (Fig.5). HSF1 ablation did not affect propagation of primary normal human CD34 HSPC but impaired propagation of human LSCs from three patients in NSGS mice. Interestingly, mouse and human MLLr AML cells were sensitive to a small molecule functional HSF1 inhibitor (Fig.6). Finally, immunostaining of primary BM sections suggests that HSF1 could be a general marker of malignancy (Fig.7). Overall, this is a very interesting paper rich of data, well-written and easy to follow. Although some links between HSF1 and AML have been raised earlier, the majority of the findings is truly novel. In addition, there might even some links for clinical translation of the molecular observations. Nevertheless, I have several, though mostly minor points that could further improve the manuscript.

Major points:

1. A large number of in vitro and in vivo experiments using rMLL-AF9 or human MLLr AML cells illustrate the effects of HSF1 for induction and maintenance of the transformed state. Some effects were also shown in human primary AML cells. One wonders, how dependent are the observed effects on the presence of an MLL-rearrangement? HSF1 seems to be generally higher expressed in AML, and mostly in cases of complex karyotypes: are the observed defects eventually linked to genetically unstable situation? Older studies suggested that a MLLr impairs the function of TP53. In fact, there is some recent data (<https://doi.org/10.1038/s41467-021-24064-1>) suggesting that TP53 controls HSF1 expression. The authors could discuss and speculate.

Although we tested the function of HSF1 in AML mainly using a mouse MLL-AF9 AML model and human MLL-rearranged AML cell lines, based on a number of observations we believe the

role of HSF1 in AML is not limited to those with MLL rearrangements. First, the immunohistochemical staining of HSF1 on human AML samples, which includes a variety of AML subtypes, showed that HSF1 is highly expressed in all AML samples regardless of the presence or absence of MLL rearrangement. Second, we tested the role of HSF1 ablation using non MLL-rearranged human cell lines (blast crisis of CML – AML equivalent, cell line K562 (not shown) and AML cell line Kasumi-1, which also showed reduced cell proliferation (**Figure S5B**). Third, and importantly, among the six primary human AML samples used for xenotransplantation in this study, only one is MLL-rearranged (**Supplemental Table 3**).

Accordingly, on page 22, line 5, we added “It should be noted that the role of HSF1 in LSC maintenance is not limited to AML with MLL rearrangement, as among the six primary AML samples used for xenotransplantation experiments, only one is MLL-rearranged. The generalized HSF1 nuclear positivity in all tested AML samples, regardless of associated cytogenetic/molecular mutations, further support a broader role of HSF1 in AML.”.

We thank the reviewer for raising the good point that TP53 controls HSF1 expression. Actually, TP53 is frequently mutated in AML with complex karyotype (*Weinberg et al., 2022 Blood Adv and Rucker et al., 2012 Blood*), which may account for one of the reasons that HSF1 mRNA expression is relatively higher in AML with complex karyotype (**Figure S4B-C**) and is consistent with that mutant cancer genome, which induces genetic instability, is one of the causes upregulates the expression of HSF1 (*Toma-Jonic A, 2019*).

Accordingly, on page 25, line 10, we added “It should be noted that TP53 is frequently mutated in AML with complex karyotype, which may account for relatively higher HSF1 mRNA expression in AML with complex karyotype (**Figure S4B-C**) given that wild type TP53 suppresses HSF1 expression and the mutant cancer genome.” and on page 25 line we included reference suggested by the reviewer to negative regulation of HSF1 by p53 as one potential source of HSF1 activation in AML.

Minor points:

1. Line 130: “no studies have been conducted to...”, this is in part correct, however, that genetic HSF1 inactivation reduced clonogenic activity of putative cancer stem cells has been previously shown (Newman B, et al., *Cancer Research*, 72:4551-4561, 2012). Maybe you adapt the sentence and add this reference?

We modified the sentence on page 7, line 6 as “few studies have been conducted to specifically evaluate the role and mechanism of action of HSF1 in AML stem cells.”, and cited the paper by Newman B et al.

2. The authors showed that Hsf1 ablation reduced the frequencies of putative LSC associated with cellular differentiation (Figure 1H-I): did they also check for their clonogenic potential? Will there be more differentiated colonies? How are the cells look morphologically upon Hsf1 inactivation?

The impact on colony formation assay in the absence of HSF1 has been shown in **Figure 1D-E**. In response to the reviewer's suggestion, we repeated the colony formation assay again using sorted LSCs and collected cells from methylcellulose cultures and performed Wright-Giemsa staining, which showed that there are more mature monocytic leukemia cells in HSF1-ablated cultures compared with vehicle treated cultures. The results are presented in the revised manuscript as new **Figure 1J**.

3. Fig.1J is not clear: in the text it says that... when FACS-sorted LSCs from Tam-treated or vehicle treated (Con) leukemic mice were transplanted into..., but the figure only contains 2 lines, +/-TAM? Which population is this now?

In the revised manuscript **Figure 1J** is now **Figure 1K**. In the figure, the blue line indicates mice receiving LSCs from full-blown MLL-AF9 HSF1^{fl/fl}creER leukemic mice treated with TAM, and green line indicates mice receiving LSCs from full-blown MLL-AF9 HSF1^{fl/fl}creER leukemic mice without TAM treatment. To make it clear, in the revised manuscript we now label the blue line as "HSF1-ablated LSCs" and the green line as Vehicle-treated LSCs (Con LSCs).

4. In Fig. 1K (now **Figure 1L** in the revised manuscript) they show that repetitive TAM application abrogated leukemic induction in primary transplanted mice: was the same observed in secondary transplants?

In response to the reviewer's suggestion, we performed the secondary transplantation experiment using whole BM cells from repetitive TAM treated or untreated MLL-AF9 HSF1^{fl/fl}creER leukemic mice (frozen cells from original Figure 1K) and show the new data as **Figure S1E** in the revised manuscript, which shows that repetitive Tam treated leukemia cells also lose the engraftment capacity in the secondary transplanted mice.

5. They explored the impact of Hsf1 ablation on normal hematopoiesis: in which mice and at which age was this done: Vav-iCre or CreER? The first is constitutive starting around E12.5; any effects might be compensated, whereas the other would allow to show the imminent effects?

We used HSF1^{fl/fl}Vav-cre mice to investigate the impact of HSF1 ablation on normal hematopoiesis. To clarify this, on page 10, line 7, we added "using HSF1^{fl/fl}Vav-cre mice."

To investigate the impact of acute HSF1 deletion on normal hematopoiesis, we have performed two additional BM transplantation experiments in response to the reviewer's suggestion: 1) treated HSF1^{fl/fl}creERT mice with Tam first, then transplanted whole BM cells one week post treatment, and 2) transplanted whole BM cells from HSF1^{fl/fl}creERT mice first, then treated with Tam or vehicle 8 weeks post transplantation and checked the donor cell chimerisms two weeks after the last dose of Tam. We found that acute HSF1 deletion by Tam, similar to HSF1 ablation by Vav-Cre, also does not have significant impact on normal hematopoiesis. These results are presented as **Figure S2F and S2G** in the revised manuscript.

On page 11, line 4, we added "In addition, we found there were no significant impacts in donor chimerisms when HSF1 was acutely ablated in adult hematopoiesis by Tam treatment (**Figure S2F and S2G**)."

6. To activate Cre-ER *in vitro* the authors used 4-OHT, while *in vivo* they used TAM, why?

The reason we used 4-OHT for *in vitro* experiments and TAM for *in vivo* experiments is because TAM can be converted into 4-OHT (an active metabolite of TAM) in the mouse liver but cannot *in vitro*. In addition, 4-OHT binds to estrogen receptor ~100-fold more efficiently than tamoxifen (Johnson MD, Zuo H, Lee KH, Trebley JP, Rae JM, Weatherman RV, et al. Pharmacological characterization of 4-hydroxy-N-desmethyl tamoxifen, a novel active metabolite of tamoxifen. *Breast Cancer Res Treat.* 2004;85:151–159.), but is more expensive.

On page 39, line 9 in the revised manuscript we added “The reason we used 4-OHT for *in vitro* experiments and TAM for *in vivo* experiments is because TAM can be converted into 4-OHT in the mouse liver but cannot *in vitro*. In addition, 4-OHT binds to estrogen receptor ~100-fold more efficiently than tamoxifen but is more expensive.”.

7. They studied HSF1 transcriptional targets in HSFKO vs control LSC: this is not clear: when were the cells taken, how many mice were studied? Was this from inducible or constitutively HSF1KO mice?

The LSCs used for RNA-seq analyses were isolated from two full blown leukemia mice developed from MLL-AF9-transduced HSF1^{fl/fl}Vav-Cre HSPCs and two full blown leukemia mice developed from MLL-AF9-transduced HSF1^{fl/fl} HSPCs (duplicate per genotype), respectively. We clarified the related description under **RNA-Seq and Transcriptome Analysis** section on page 36, line 9 in the revised manuscript, and added “two full blown leukemia mice developed from MLL-AF9-transduced HSF1^{fl/fl}Vav-Cre HSPCs and two full blown leukemia mice developed from MLL-AF9-transduced HSF1^{fl/fl} HSPCs (duplicate per genotype), respectively.”.

8. Transcriptome and ChIP-seq raw data needs to be posed into an open repository (like GEO): here the data is presented in a very selective manner.

We have already deposited the RNA-seq data and CUT&RUN data in the National Center for Biotechnology Information’s Sequence Read Archive (SRA) database with the BioProject ID: PRJNA782637. In addition, the RNA-seq data are also presented as **Supplemental Table 1** in the revised manuscript.

9. HSF1 knockdown in primary AML (a-c) was performed (Fig.6G). What was the efficacy of the KD? In addition, we do not know anything about the nature of these primary AML samples: do they carry MLLr, and if so, which ones?

The knockdown efficacy is about between 70-90%. Accordingly, on page 18, line 8 from bottom in the revised manuscript, we added “70-90% of knockdown efficacy, **Figure S5A**.”.

The limited information on these de-identified primary AML samples used for xenotransplantation is now shown in Supplemental Table 3 in the revised manuscript.

10. Line 391-394: In a situation of no detectable blasts, some residual HSF1-positive remained after chemotherapy: what kind of cells are those then? Very immature LSCs?

The nature of these residual HSF1 positive leukemia cells is currently unknown. In some cases, these remaining HSF1-positive leukemia cells disappeared during follow up, and in others the number of these cells were increased. Whether they are residual LSCs with altered immunophenotype or metabolic status is unknown, which needs to be investigated further. Accordingly, on page 20, line 5 from bottom, we added “although we do not know whether they are residual LSCs with altered immunophenotype or metabolic status due to treatment”.

11. In the reviewer’s opinion, the data of Fig.7 appears rather too preliminary to make any strong general statement, so the authors could eventually implement some of the data in a small panel in Fig.6 (and eventually follow-up these findings in more in-depth study?).

We agree and understand the reviewer’s consideration. We simplified Figure 7 and presented them as **Figure 6N to 6S** in the revised manuscript and we have softened our interpretation of the data.

Reviewer #2:

General comments:

It is generally accepted that leukaemic stem cells (LSCs) in acute myeloid leukaemia (AML), and other stem cell driven leukaemias, are of significant clinical importance. In this manuscript the authors describe the role of Heat Shock Transcription Factor 1 (HSF1), a central regulator of the heat-shock response, in AML, using both in vitro and in vivo approaches. The paper is well written and clearly presented. The data are robust in most parts and the topic is of interest to the reader of Nature Communication.

Initially, using conditional HSF1 deletion mice and MLL-AF9–induced AML mouse model the authors demonstrate that genetic deletion of HSF1 impairs leukaemogenesis and LSC function/self-renewal, without significantly affected normal haematopoiesis. Experimental data in this part are convincing.

By performing RNA-sequencing on HSF1 knockout cells, the authors show that HSF1 ablation led to broad gene deregulation, including downregulation of genes involved in the tricarboxylic acid (TCA) cycle and oxidative phosphorylation (OXPHOS), with further work pointing towards downregulation in complex II activity. This part is also convincing, although additional work is needed to strengthen this part (see specific comments below).

Next the authors show that higher HSF1 expression correlates with a poor prognosis in AML and HSF1 knockdown impairs transplantation of human AML cells, with minimum effect on transplantation of normal (CD34+) stem/progenitor cells. Encouraging data was also obtained using pharmacological inhibition HSF1 (using of SISU-102, a previously described HSF1

inhibitor).

Overall, the authors use robust models of AML and present convincing data regarding critical role of HSF1 in leukaemia. The mechanistic data hints that a contributing factor is downregulation of OXPHOS, which is interesting, although additional experiments need to strengthen this part.

Specific comments:

The authors define the AML LCS population in the MLL-AF9 model as lin-ckit+Sca1-CD16+CD34+ (according to Ref 33) and refer to “LSCs” throughout the text. However, it is not clear how enriched this (granulocyte macrophage progenitors-like?) population is for LSCs, which may have implication when this population is used in in vitro assays, for example in Figures 4-5. Please clarify in text. Also, can the authors experimentally test whether MLL-AF9 expression drives increased OXPHOS in this (or other relevant LCS-enriched) cell population (compared with non-transduced cells)?

The LSC (lin-ckit+Sca1-CD16+CD34+) in mouse MLL-AF9 AML model has been defined and functionally tested previously (Ref#33, 2006 *Nature* **442**, 818-822), and showed enhanced leukemogenesis compared with non-LSCs. In the original submission, in addition to LSC, we also used the term LSPC (leukemia stem/progenitor cells) (original line 235-236) which has been defined as CD11b⁺cKit^{high} AML cells (2015, *J Clin Invest* **125**, 1286-1298) because the cultured leukemia cells rapidly lost lineage negative population. In response to the reviewer’s suggestion, we have now clearly labeled the population (LSCs or LSPCs) used for each experiment in the text and figure legends of the revised manuscript.

As suggested by the reviewer, we isolated CD11b+Kit+ LSPCs and non-transduced cKit+Lin- bone marrow cells and measured OXPHOS using Seahorse and found that MLL-AF9-transduced cells significantly increased OXPHOS compared to un-transduced normal bone marrow Kit+Lin- cells (**Figure S3A** in the revised manuscript). On page 15, line 5, we added “We found that in the murine MLL-AF9 AML model, MLL-AF9 transduced cKit⁺Lin⁻ cells had higher OXPHOS activity than non-transduced BM cKit⁺Lin⁻ cells (**Figure S3A**).”.

Figure 4, D-F: “Con” and “HSFKO” legends on the graph is confusing (wrong?) – there is only one differentially expressed gene list, which is compared with a gene set, so not sure it’s accurate to place these two legends on the graph.

Compared with control LSCs (Con), the HSF1-ablated LSCs (HSF1KO) showed an impaired stem cell signature (**D**), down-regulated genes related to OXPHOS (**E**) and TCA cycle (**F**). We think that it can make it clearer with labeling than without labeling. If the reviewer thinks this is still confusing, we can delete the labeling.

Figure 5, J: Why is glucose only traced for 60min? What is the relative contribution of labelled carbons to each metabolite after such a short labelling period? I would suggest the authors also show the labelling fraction of each isotopologue (m+1....m+6) for the selected metabolites as

supplementary data.

Sixty minutes tracing allows for adequate detection of relevant ¹³C isotopologues, while minimizing the amount of label scrambling that occurs independent of net flux. Glucose enters the cell and labels glycolytic intermediates within a few minutes, and labeling of TCA cycle intermediates can take an hour or more, depending on the cell type and metabolic rate. However, at overly long time points, ¹³C-labeled TCA cycle metabolites can cumulatively be exported to the cytoplasm, metabolized, and then reenter the TCA cycle, leading to isotopic labeling patterns that are increasingly difficult to interpret. Therefore, it is ideal to trace at a time point that is long enough for glucose to label glycolysis and the TCA cycle above the limit of quantitation but not so long that label mixing confounds interpretation.

Accordingly, on page 34, line 1, in the revised manuscript we added “when it will be long enough for glucose to label glycolysis and the TCA cycle above the limit of quantitation but not so long that label mixing confounds interpretation.”.

As suggested by the reviewer, we also showed the labelling fraction of each isotopologue for the selected metabolites in **Supplemental Table 2** and added “**and Supplemental Table 2**” on page 16 line 1 in the revised manuscript.

Figure 5: Can the authors speculate why there is no compensation by increased glycolysis if the main effect of HSF1 ablation is reduction in Complex II activity?

Because LSCs are primarily dependent on mitochondrial oxidative phosphorylation for ATP, the reduction of Complex II activity by HSF1 ablation may be induction of cell sickness and metabolic dysfunction that is too great for the cells to overcome. LSCs with HSF1 ablation show decreased proliferation (**Figure 3D**), increased ROS production (**Figure 5D-E**), and increased apoptosis (**Figure 3E**), suggesting that the cells are in crisis and potentially unable to alter glycolysis to compensate for ETC and TCA cycle dysfunction. In addition, HSF1 exerts its impact on LSPC metabolism at multiple levels, as HSF1 inhibition also inhibits the expression of HSF1 target genes related to glycolysis (**Figure S3B**).

Accordingly, on page 15, line 10 in the revised manuscript, we discussed this as “The apparent lack of compensation of ATP production by increased glycolysis in HSF1-ablated LSPCs suggest that LSPCs are in crisis – decreased proliferation (**Figure 3D**), increased ROS production (**Figure 5D-E**), and increased apoptosis (**Figure 3E**). The inability to alter glycolysis to compensate for ETC and TCA cycle dysfunction could also be due to HSF1 ablation impairing the expression of genes related to glycolysis (**Figure S3B**).”.

Figure 5: Does overexpression of SDHC rescue OCR? Does overexpression of SDHC rescue Complex II activity?

In response to this excellent question, we performed Seahorse experiments and measured the Complex II activities using vector or SDHC overexpressing LSPCs in the presence or absence of 4-OHT. As expected, overexpression of SDHC significantly restored the impaired OCR and the

ETC Complex II activities induced by HSF1 ablation. These results are presented as **Figure 5Q and 5R**, respectively in the revised manuscript.

On page 17, line 2, we added “In addition, overexpression of SDHC significantly restored the impaired OXPHOS and the ETC complex II activities (**Figure 5Q and 5R**).”.

Figure 6: Please clarify whether engraftment of human cells is measured in blood or bone marrow? Why is there so low transplantation inefficacy of human CD34 (how many cells transplanted?). Do mice transplanted with human AML cells die of leukaemia? Why do authors only rely on engraftment but not survival of mice?

All the engraftment of human cells in **Figure 6** is measured using bone marrow cells after the mice were sacrificed. To clarify this, in the revised manuscript on page 40, line 5, we added “using BM cells”.

The low transplantation efficiency of human CD34⁺ cells was most likely due to the quality of the samples. We enriched CD34⁺ cells from the leftover negative lymphoma staging bone marrow aspirates. To address this point, we repeated this experiment using the purchased CD34⁺ BM cells (Stemcell Technologies) and represented the data as a new **Figure 6B** in the revised manuscript.

Termination of the transplantation experiments at the same time enabled us to analyze the impact of HSF1 inhibition on engraftment. Per the reviewer’s suggestion, we also performed another two sets of transplants and monitored the survival of mice transplanted with genetically HSF1 deleted human AML cells (**Figure S5C**) or transplanted with human AML cells and then treated the mice with a small molecule HSF1 inhibitor (**Figure S5D**). In addition, we also tested combination therapy using the HSF1 inhibitor and Ara-C compared with single agent alone (**Figure S5E**).

Accordingly, in the revised manuscript we added “and extended the survival of recipient” on page 18, line 5 from bottom, and page 19, line 9 from bottom, respectively. We also added “Furthermore, the combination of SISU-102 with Ara-C further suppressed the engraftment of human AML cells (**Figure S5E**).” on page 19, line 8 from bottom.

For in vivo experiments, why are cells transplanted via retro-orbital venous sinuses instead of tail vein injection?

Our experience is that for normal HSPC transplantation, both retro-orbital and tail vein injection work equally well. For leukemia cell transplantation, tail vein injection is better because occasionally the residual AML cells can cause localized tumor formation in the orbital region.

Accordingly, on page 38, line 1, we added “In our experience, both retro-orbital and tail vein injection work equally well for normal HSPC transplantation. For leukemia cell transplantation (see below), we used tail vein injection to avoid the occasional localized tumor formation in the orbital region caused by the residual AML cells.”.

Reviewer #3:

A fundamental problem in cancer therapy in general, and acute myeloid leukemia (AML) in particular, is that although during the conventional therapy, the malignant cells are killed, specific self-renewing cancer stem cells can survive. Accordingly, AML has been shown to be maintained by leukemic stem cells (LSCs), which have a capacity to re-initiate the progression of AML. The goal is to understand the regulatory mechanisms keeping LSCs alive and supporting their self-renewal. In this study, conventional and conditional HSF1 knockout mice were utilized to show that this transcription factor is required for the initiation and maintenance of murine AML, while sparing the steady-state and stressed hematopoiesis. The depletion of HSF1 was shown to dysregulate genes that are involved in modulating LSC stemness and the heat shock response to suppress the mitochondrial oxidative phosphorylation via reduction of enzymes, SDHC and SDHA, that regulate the mitochondrial respiratory chain complex II. SDHC is a direct target of HSF1 and by forcing the expression of SDHC, AML developmental defects caused by HSF1 gene repression were restored. The growth and engraftment of human AML cell lines and primary AML cells were shown to be reduced by HSF1 knockdown or in response to administration of a direct HSF1 small molecule inhibitor (SISU-102). Finally, the expression of nuclear HSF1 was shown to correlate with the AML status of patients and was proposed as a potential clinical marker to monitor the status and progression of AML.

The topic of the study is very timely and important. The obtained results are mostly of high quality, clearly presented and well controlled (for specific comments, see below). The manuscript is very well written, the methods used are many and in good balance, including appropriate mouse and cell models in combination with human patient material. The authors could give some more insight into the mechanism by which the HSF1 inhibitor could specifically be targeted to LSCs so that it would not affect healthy cells. Although they do show that HSF1 depletion does not affect normal hematopoiesis and that normal cells do not express high levels of HSF1, it would be important to show the effect of this inhibitor (SISU-102) on other cell lines as well. Since inactivation of HSF1 has been proposed as a potent treatment for AML already earlier, for example, by targeting HSF1 with specific miRNAs and by inhibiting eIF4a, the authors should discuss in more detail the benefit(s) of using SISU-102 over the previously presented targeting strategies. It would be interesting to know what upregulates HSF1 expression in AML and whether high HSF1 is a driver of the AML phenotype.

We greatly appreciate the reviewer's positive comments and invaluable suggestions.

In fact, the effects of SISU-102 on other cells *in vivo* have been reported by our collaborator Dr. Thiele's group (Ref#52 *Sci Transl Med* 2020 Dec 16;12(574): eabb5647). Specifically, during the 3-week course of treatment, daily SISU-102 administration elicited no observable adverse effects on mouse behavior, weight, or organ histopathology using levels of SISU-102 five times that used in the current work (**Figure 6D and S10** in the above paper). Moreover, SISU-102 inhibited the proliferation of prostate cancer cells with a much greater potency as compared to benign prostate cells.

Accordingly, on page 19, line 6 of the revised manuscript, we added “Importantly, SISU-102 administration elicited no observable adverse effects on mouse behavior, weight, or organ histopathology.”.

Despite genetic validation of HSF1 as a therapeutic target in a range of cancers and HSF1 pathway inhibitors being identified and evaluated in cellular and mouse xenograft cancer models, these inhibitors either act in an indirect manner in the HSF1 pathway or have an unknown mechanism of action. SISU-102 is a direct and selective small-molecule HSF1 inhibitor, which physically engages HSF1 and selectively stimulates degradation of nuclear HSF1.

On page 18, line 2 from bottom, we added “Previously reported HSF1 pathway inhibitors have been evaluated in cellular and mouse xenograft cancer models. However, most either act in an indirect manner in the HSF1 pathway or have an unknown mechanism of action. SISU-102 (recently published as DTHIB, Direct Targeted HSF1 InhiBitor) has been shown to be a direct and selective small-molecule HSF1 inhibitor which physically engages HSF1 and selectively stimulates degradation of the active, homotrimeric, nuclear form of HSF1.”.

In response to the reviewer’s question on what upregulates HSF1 expression in AML we have modified the manuscript to clarify and speculate on this point as follows on page 24 lines 4 from bottom:

“Among the mechanisms that result in increased levels of nuclear HSF1, in a broad range of cancers, include amplification of the HSF1 gene located within chromosome 8q24.3, reduced expression of the HSF1 degradation F box E3 ligase Fbxw7, activation by loss of p53 and increased HSF1 transcription by oncogenic signaling pathways such as NOTCH signaling. Other potential contributing factors are the accumulation of the mutant cancer proteome and rapid proliferation of cancer cells, which may select for increased levels and activity of HSF1 via additional mechanisms.”.

We suggest that AML stem cells require the functional HSF1 for self-renewal, but HSF1 itself is not an oncogene, not a driver of the AML phenotype.

The reviewer’s point about other proposed therapeutic approaches proposed for AML that include miRNA and eIF4a inhibitors is excellent and does beg for a comparison to the use of HSF1 inhibitors such as SISU-102. As SISU-102 directly binds and inhibits HSF1, the propensity for “off pathway” negative consequences such as might be expected for inhibitors of the eIF4a helicase, involved in the initiation of protein translation, would be minimized. Moreover, the use of small molecule HSF1 inhibitors could have significant absorption and other pharmacokinetic advantages over the use of microRNAs. We have added a brief discussion of this point on page 23, line 5 from bottom, as “Previous reports suggest targeting HSF1 in AML through microRNAs or via inhibition of translation initiation factor eIF4a. However, directly targeting HSF1 with small molecules such as SISU-102 may have advantages with respect to absorption and pharmacokinetic features and would limit potential toxicity from “off target” pathway modulation.” in the Discussion section of the revised manuscript.

Specific comments:

1. Lines 99-101: The authors are not expected to provide detailed description of all the AML subtypes, but it could be useful to state that the WHO classification divides AML into distinct categories based on the genetic alterations and immunophenotypic markers. The intriguing part is that the stalling of the blast differentiation can be caused by various alterations resulting in diseases heterogeneity. Perhaps it would make the point even stronger, regardless of the subtype, the LSCs are behind the disease in every case.

We thank the reviewer for this suggestion. As the reviewer suggested, on page 5, line 7, we added “The WHO classification divides AML into distinct categories based on the recurrent cytogenetic abnormalities, molecular mutations, morphology and immunophenotypic features. The aberrant leukemia blast proliferation and the stalling of blast differentiation are common features of AML. Regardless of the subtype, all....”.

After that it would be better to start the more detailed description about LSCs (Line 104), which could be extended to cover also other details additional to the mitochondrial specialties. For example, information about the CD-molecules and other markers that are used later on would be really helpful. What markers are specific for the LSCs? What distinguishes the LSCs from other leukemic cells?

In the revised manuscript on page 5, line 6 from bottom, we added “AML stem cells are self-renewing quiescent blasts. Depending on the subtype, some LSCs are positive for CD34, like its normal counterpart, but others, for example, AML with NPM1 mutations and AML with monocytic differentiation, are negative for CD34. As there is no general surface marker for LSCs, AML stem cells have recently been enriched through their low reactive oxygen species (ROS) production status compared with the bulk blasts.”.

2. Line 108: Please add reference to the sentence describing the OXPHOS.

We have added references accordingly.

3. Line 135: Clinical markers are not really malignant, and AML is already malignant. Rephrase: nuclear HSF1 levels may be used as a clinical marker to monitor AML status.

In the revised manuscript on page 7, line 11, we deleted “malignant” and rephrased the sentence as “nuclear HSF1 levels may be used as a clinical marker to monitor AML status”.

4. Suggestion for the conclusions on line 134 onwards: The authors could elaborate more on these ideas by thinking about the possibility for companion diagnostics (e.g. similar to PD-L1 and Keytruda, HER2 and Trastuzumab etc.). HSF1 IHC together with SISU-102 as a CDx would be a useful strategy, which could also be added to the Discussion.

We thank the reviewer for the insightful suggestion. In the revised manuscript on page 24, line 15, we added “In addition, the combination of monitoring HSF1 expression by immunostaining

along with HSF1 small molecular inhibitors may be a useful strategy to target AML residual disease just as done for PD-L1 expression with Keytruda and HER2 expression with Trastuzumab in the recently described companion diagnostics.”.

5. Line 146: Albeit only HSF1 is in the focus of this study, mammals do have also other HSFs and therefore, the name of the KO mouse should be changed to HSF1KO so that the readers do not accidentally think that you have knocked out all murine HSFs.

We have now clearly labeled as HSF1KO or HSF1^{fl/fl} throughout the revised manuscript and figures.

6. Line 146: The sentence related to Figure 1A. The figure indicates the survival difference of the mouse lines, but not exactly the differences in the leukemogenesis. Please consider rephrasing.

On page 8, line 8 of the revised manuscript, we rephrased the sentence as “the development of MLL-AF9 leukemia *in vivo* was significantly delayed”.

7. Is the WB for Figure 1C, D, E also? Rephrase or add the information of successful downregulation of HSF1.

We confirmed the deletion of HSF1 protein by Western blot and showed the results as **Figure S1B** on page 8, line 4 from bottom.

8. Line 172: Correct the typo with the CD34+33 and check the superscripts throughout the manuscript.

33 is the reference number (now Reference #35). We now changed “CD34+33” to (lin⁻ckit⁺Sca1⁻CD16⁺CD34⁺).

9. Line 173: Describe the meaning of these markers. CD11b positivity indicates a state where the cell has progressed with specific differentiation trajectory or? The same comment concerns Line 459.

On page 9, line 9 in the revised manuscript, we added “The LSCs in MLL-AF9-induced AML have been defined as lineage marker-negative, cKit-positive, Sca1-negative, CD16-positive, and CD34-positive (defined as lin⁻ckit⁺Sca1⁻CD16⁺CD34⁺) granulocyte-macrophage progenitor-like cells.”.

On page 9, line 14, we added “more mature and differentiated” to explain CD11b.

10. Figure S2C is missing (Line 190) due to a mislabeling of the panels in Figure S2. Please, correct.

Thanks for spotting this error. In the revised manuscript we have now fixed the labels in **Figure S2** (D to C, E to D and F to E).

11. Line 234: What is the difference between LSCs and LSPCs? If none, please harmonize. Also, the previous text gives an impression that CD11b is a marker for more differentiated state. Here it sounds it would be used as a stem cell marker. Please, clarify.

In the revised manuscript, we defined LSCs in MLL-AF9-induced AML as $\text{lin}^{-}\text{ckit}^{+}\text{Sca1}^{-}\text{CD16}^{+}\text{CD34}^{+}$ (2006, *Nature* **442**, 818-822) and LSPC (leukemia stem/progenitor cells) as $\text{CD11b}^{+}\text{cKit}^{\text{high}^{+}}$ AML cells (line 235-236) (2015, *J Clin Invest* **125**, 1286-1298). We have now clearly labeled the population (LSCs or LSPCs) used for each experiment in the text and figure legends (See also the response for Reviewer #2's first specific comments).

12. Figure 4A-C contain two columns for Con and HSFKO. Is the data in the columns derived from two different mice? This should be indicated. Figure 4 shows the effect of HSF1 depletion on the expression of selected target genes. It would be interesting to see the total number of genes that are affected by HSF1 depletion, which could be added as a supplementary data.

We added “isolated from 2 full blown individual leukemia mice for each genotype” in the Figure legend of **Figure 4**.

In the revised manuscript we provided the gene expression profiles as **Supplementary Table 1**.

13. Line 266: The authors claim that CUT&RUN identifies HSF1-regulated genes, although the method only detects HSF1-bound sequences. For example, in HS, a subset of HSF1-bound genes is not differentially expressed upon heat shock. Please, correct.

We agree with the reviewer, and on page 13, line 5 from bottom of the revised manuscript, we changed it to “HSF1-bound” to make it more accurate.

14. In Figure 5C–E, the labels indicating what is measured in Y-axis of the bar graphs are missing. The two flow cytometry plots are presumably from control mice and 4-OHT treated mice, which should be indicated.

In the revised manuscript we fixed the labeling for Figure 5C-E, and they are clearer now.

Four compounds, including oligomycin, antimycin A, rotenone, and FCCP, are shown in Figure 5F–G. However, the main text does not mention anything about these compounds. Following is mentioned in the Materials and Methods: “For quantification of ATP production rate from both glycolytic and mitochondrial pathways and ATP rate index, analyses were performed with injection of 15 μM oligomycin (Sigma-Aldrich, Cat# 871744), 5 μM antimycin A (Sigma-Aldrich, Cat# A8774), and 5 μM rotenone (Sigma-Aldrich, Cat# R8875).”

What do these compounds cause in cells? Please, clarify in the main text.

We have now clarified these compounds in the main text. On page 14, line 2 from bottom, we added “In this assay, the administration of oligomycin inhibits ATP synthase, resulting in a reduction in mitochondrial respiration or OCR. The injection of Carbonyl cyanide-4

(trifluoromethoxy) phenylhydrazone (FCCP) treatment will enhance electron flow through the electron transport chain (ETC) and drive maximal oxygen consumption by complex IV. A mixture of rotenone (a complex I inhibitor) and antimycin A (a complex III inhibitor) will shut down mitochondrial respiration and enable the determination of the contribution of nonmitochondrial respiration”.

Lines 316-319: “As succinate is the metabolic substrate and fumarate is the product for ETC II decreased fumarate and malate levels suggested a defect in ETC complex II activity (Figure 5I-J). Consistent with this prediction, ETC II activity was markedly decreased in HSF1-ablated LSCs compared with control LSCs (Figure 5M).” What method was used to test this?

We have now clarified the assay information in the Methods section of the revised manuscript. Accordingly, on page 29, line 9 from bottom, under Electron transport chain complex II activity assay, we added “...by the colorimetric Complex II Enzyme Activity Microplate Assay Kit”.

Figure legend for Figure 5I mentions n=6, but there are only five columns. Are there five or six replicates? Figure legend for Figure 5K is missing. In Figure 5L, red indicates control and blue indicates HSKO. This must be an error, since the data suggest that HSF1 depletion leads to increased expression of SHD A and SDH C, although the opposite is indicated in the text.

Thanks for spotting these mistakes, which have been corrected in the revised manuscript.

15. Line 344: It is somewhat confusing what cell types are compared in the figure. Is there a difference in HSF1 mRNA and protein expression levels between normal hematopoietic stem and progenitor cells and LSCs? Is there a difference in HSF1 mRNA and protein expression levels between non-stem cell populations in disease and health states? The authors want to emphasize the role of HSF1 in the self-renewing population, but now it remains unclear what populations are compared in Figure 6C.

Figure 6C showed the difference in HSF1 mRNA expression in Human bone marrow CD34⁺ hematopoietic stem/progenitor cells vs CD34⁺ leukemia blasts. There is no difference in HSF1 protein expression between non-stem cell populations and stem cell populations in AML, given the diffuse HSF1 positivity in all AML samples by HSF1 immunostaining. However, all AML cells express higher levels of HSF1 protein than those essentially negative control bone marrow cells (**Figure 6D, E, S4A**).

To make this clear, on page 17, line 1 from bottom, in the revised manuscript, we added “In addition, there is no difference in HSF1 protein expression between non-stem cell populations and stem cell populations in AML given the diffuse nuclear HSF1 positivity in all AML samples by HSF1 immunostaining (**Figure 6E and S6B**).”.

Lines 349-350: “[...] HSF1 protein is highly expressed in human leukemia cells, with an overall high level in the nucleus.”. This refers to Figure 6E, but the figure is of low quality, and the low contrast between HSF1 in red and CD34 in brown makes it difficult to determine if there is high nuclear localization. Presenting results from the quantification that was done according to the methods section would resolve this issue.

The HSF1 expression is in the nucleus in all the human AML cells. In the revised manuscript we now have provided the high-resolution pictures for **Figure 6E**. In addition, we provided a high-resolution of HSF1 immunostaining picture in **Figure S6B**, which clearly shows the nuclear staining pattern in human AML cells.

Lines 381-383: "[...] diffuse positivity of nuclear HSF1 in AML BMs [...]". This refers to Figure 6E and has the same issue as above. Is there counterstaining in Figure 6E? Please add scale bars. Close-ups would also be helpful. How many patients were analyzed for Figure 6F?

See above response related to the reviewer's comment on **Figure 6E**. We also added scale bars.

For the survival curve (**Figure 6F**), which is based on TCGA data, there are 34 AML patients in the HSF1^{high} group, and 133 AML patients in the HSF1^{low} group. Accordingly, we updated **Figure 6F** to include the above information.

Figure 6A: The title for the Y-axis is "expression", which is not descriptive enough. Even if it is explained in the figure legend "mRNA expression" and HSF1 as the histogram title would be clearer. Be consistent with the style how to present the data, for example, in Figure 5L the Y axis title is "Relative mRNA levels".

We have now labeled the Y-axis of **Figure 6A and 6C** as "Relative mRNA level" to make it consistent throughout the revised manuscript.

"As shown in Figure 6I, SISU-102 clearly inhibited the expression of the HSF1 targets, HSP90 and SDHC, in a dose-dependent manner and strongly suppressed the growth and OCRs of murine MLL-AF9 and human MV4-11 cells (Figure 6J-K)."

Fig. 6I-K show three cell lines, HSF^{fl/fl}creER, MV4-11, and NOMO-1. NOMO-1 is mentioned only in the Materials and Methods, but it should be mentioned also in the main text. It would be also good to remind the reader that HSF^{fl/fl}creER is a modified MM-AF9 cell line, since the text for Fig. 6I-K mentions only MLL-AF9.

We have made the description clearer and on page 19, line 12 from bottom of the revised manuscript, we added "HSF1^{fl/fl}creER leukemia cells" and "NOMO-1 AML", respectively.

HSF1 levels in response to SISU-102 should be shown in Figure 6I.

We have provided the expression of HSF1 protein in **Figure 6I** and modified the related figure legend. In addition, on page 19, line 10, we added "and HSF1 itself".

16. The results associated with Figure 7 are difficult to comprehend and follow. The results from patient materials (Lines 384-396, Figure 7) mention HSF1 levels in general, but L396-397 suddenly mentions that only HSF1 nuclear expression can be used to monitor AML status. Figure 7A and S4 has "HSF1 H-index" on the Y-axis but the corresponding figure legends calls it "H-score index". Based on the Methods section (Lines 707-714), this should simply be "H-

score", and it should be mentioned in the paper what the score is based on (eg. whole cell HSF1 signal or nuclear only HSF1 signal). Based on the Materials section, it seems like the H-score was calculated for nuclear signal. Additionally, Lines 713-714 states "The intensity score, proportion score, H-score and positivity index were reported", but aside from the H-score box plots in Figures 7A and S4, they are not reported in the paper. Based on this section, it seems the authors have data from these quantifications that indicates HSF1 was nuclear, so this quantification (cytoplasmic vs. nuclear) should be included in the manuscript. The finding that nuclear HSF1 expression can be used to monitor AML status is a major result in this study, and therefore, it is important to provide as strong evidence for this statement as possible.

Thanks for detailing these issues. By immunohistochemical staining, HSF1 is mainly expressed in the nucleus of all 138 tested diagnostic AML samples. We have provided this information in the text related **Figure 6E and S6B** (see above) and reiterated the nuclear staining pattern on page 20, line 5.

We changed the label of Y-axis in **Figure 6N and S6A** to HSF1 H-score now. We also mentioned in the main text the H-Score on page 20, line 9 and added "The H-Score based on HSF1 immunohistochemical staining, calculated by Extent (%) x nuclear intensity (0, 1+, 2+, 3+) of staining".

As we only reported the H-Score, we modified the method on page 36 line 5, as "The H-score, calculated as Extent (%) x nuclear intensity (0, 1+, 2+, 3+) of HSF1 staining, was reported."

Figure 7: Please indicate scale bars either in every figure or collectively in the figure legend. The legend states that the insets in Figure 7B, D, and G are HSF1 stainings. The inset in panel D appears to be CD34 (membraneous staining pattern), perhaps there is a mix up? It would be helpful to present such FFPE sections, where the amount of the tissue is comparable. Now, especially Figure 7D and G contain only a few cells. Is there counterstaining in Figure 7E? The brightness of the images is variable.

In the original submission, we provided scale bars in **Figure 7F**; as it is not clear we have added a scale bar to **Figure 7D**. In the revised version, the scale bars are provided in each picture.

We intended to present two different relapsed AML samples, in which one has blasts positive for CD34 (patient 2, **Figure 7D-F**). In this patient, the pictures provided are CD34 (membrane) and HSF1 (nuclear) double staining. No counterstaining was performed. The brown color nuclear staining is for HSF1. There is no mix-up, just that the HSF1 staining is relatively weak. The reason why **Figure 7D and G** contains only a few cells is due to these two pictures being from remission bone marrows 2 weeks post chemotherapy (we added the information in related Figure legend), accordingly there were not many cells left. We have explained these more clearly in the revised manuscript. Note that only two cases were presented in the revised manuscript per Reviewer 1's suggestion (Reviewer 1's last comment). Note that the contents of the original **Figure 7** are presented as **Figure 6N-S** in the revision.

REVIEWERS' COMMENTS

Reviewer #1 (Remarks to the Author):

the authors nicely addressed most of the points that I raised. the MS clearly improved!

Reviewer #2 (Remarks to the Author):

The authors have addressed my comments and improved the overall quality and robustness of the work. I have no further comments.